# An Offline-Online WebGIS Android Application for Fast Data Acquisition of Landslide Hazard and Risk

Roya Olyazadeh [1], Karen Sudmeier-Rieux [1], Michel Jaboyedoff [1], Marc-Henri Derron [1], and Sanjaya Devkota [2]

[1] University of Lausanne, ISTE - Institut des Sciences de la Terre, Faculté des géosciences et de l'environnement, Lausanne, Switzerland,
[2] Department of Civil Engineering, Institute of Engineering, Tribhuvan University, Kathmandu, Nepal

*Correspondence to*: Roya Olyazadeh (Roya.Olyazadeh@unil.ch)

**Abstract.** Regional landslide assessments and mapping have been effectively pursued by research institutions, national and local governments, NGOs and different stakeholders for some time and a wide range of methodologies and technologies are proposed consequently, land-use mapping and hazard event inventories are mostly created by remote sensing data, subject to difficulties such as accessibility and terrain which need to be overcome. Likewise, landslide data acquisition for the field navigation can magnify the accuracy of database and analysis. Open-source web and mobile GIS tools can be used for improved ground-truthing of critical areas to improve the analysis of hazard patterns and triggering factors. This paper reviews the implementation and selected results of a secure mobile-map application called ROOMA (Rapid Offline-Online Mapping Application) for the rapid data collection of landslide hazard and risk. This prototype assists the quick creation of landslide inventory maps (LIMs) by collecting information on the type, feature, volume, date and patterns of landslides using open-source web-GIS technologies such as Leaflet maps, Cordova, GeoServer, PostgreSQL as the real DBMS (Database Management System) and Postgis as its plugin for spatial database management. This application comprises Leaflet map coupled with satellite images as base layer, drawing tools, geolocation (using GPS and Internet), photo mapping and events clustering. All the features and information are recorded into a GeoJSON-text file in an offline version (Android) and consequently uploaded to the online mode (using all browsers) with the availability of internet. Finally, the events can be accessed and edited after approval by an administrator and then be visualized by the general public.

Keywords: Landslide Hazard and Risk, Landslide inventory, Post Disaster, Free and Open Source Software for Geoinformatics (FOSS4G), Offline-Online Android

## 1. Introduction

Landslides refer to all types of mass movements on slopes (Varnes, 1984) and can be triggered by various external events such as intense rainfall, earthquakes, water-level changes, storm waves or human activities. The location, the time of event and the types of movements can be recorded in a Landslide Inventory Map (LIM). LIMs are important factors for hazard and risk assessments, particularly if there is a significant number of landslides with different types, dates, volumes and trigging factors (Coe et al., 2004). They can be created using various methods, however the selection of techniques depends on the size of the area, the resolution, the scale of the map, land-use, soil and geomorphology (Coe et al., 2004; Guzzetti et al., 2006; Hungr et al., 2014). Documenting landslides is essential to defining landslide susceptibility, hazard and risk and for survey types, patterns, distributions, and statistics of slope failures. However, developing complete landslide inventories is difficult, due to accessibility, the dynamic nature of landslides and also the time required (van Westen et al., 2006). Conventional techniques lead to the development of landslide inventories mainly based on the visual interpretation of satellite images, assisted by field surveys. Typical issues for creating these maps include (van Westen et al., 2006; Safaei et al., 2010; Guzzetti et al., 2012):

1. All methods for developing landslide inventories are resource intensive and time-consuming (Guzzetti et al., 2012).
2. Landslides are often small with high frequency of occurrence and located in remote areas which are difficult to access;
3. Landslides often have different characteristics which require them to be mapped and documented individually;
4. The lack of landslide documentation and databases is the main issue in the evaluation of landslide hazard risk;
5. Limited damage data are available for landslides, which are why developing landslide vulnerability assessments is challenging;
6. Sources of landslide inventories, such as aerial photography, satellite imagery, InSAR (Interferometric Synthetic Aperture Radar) and LiDAR (Light Detection and Ranging) are expensive.

Several authors have described the role of GIS for landslide susceptibility and hazards with respect to the type of data available, landslide type and potential extension (van Westen, 1993; Guzzetti, 2000; Van Den Eeckhaut et al., 2009; Carrara et al., 1991; Dhakal et al., 2000). While the above authors have noted the importance of enhanced mapping, mobile-GIS offers technology with more effective ground-truthing and a rapid tool, which can systematically fill a database, especially for inexperienced mappers. Currently, there is a high potential to apply mobile-GIS including GPS and mapping tools to significantly increase efficiencies in data collection such as location accuracy and detailed information of features.

In this paper, Rapid Offline-Online Mapping Application (ROOMA) based on Geospatial Open-Source technologies is described to collect data on landslide events, hazard impacts and damaged infrastructure, which can be made freely accessible to authorities, stakeholders and the general public. An offline technology helps to map the events, especially in rural areas where internet is not available. Besides, the preliminary result of this application is also compared to the results of satellite image interpolation. This prototype has following objectives:

1. An Android mobile application with possibility of both Offline-Online access
2. Fast and easy data and information acquisition

3. Advanced visualization using satellite images and drawing tool

4. Central database with availability by different services (mobile, PCs (Personal Computers) and standard web browsers)

5. Data management improvement in hazard event mapping and storage using new technologies such as Postgis and GeoServer.

The paper is structured as follows. In section 2, we first present the background, the importance of landslide inventories maps in hazard and risk assessment and principles of the different approaches for landslide inventory. We also review some GIS tools that simplify field navigation. Section 3 discusses the description of mapping method, with a field survey for preparation of LIMs in relation with elements at risks. Section 4 illustrates the architecture and platform using open-source geospatial technologies to map landslides by using an Android application. Section 5 and 6 focus on case study and results. Finally,

section 7 concludes by discussing the advantages of mobile-GIS, with the future outlook of producing data on landslides.

## 2.   Background

Landslide risk management estimates risk options with different levels of acceptance criteria. It includes estimations for various levels of risk, decisions on the acceptable level, recommendations and implementation of suitable control measures to reduce risk. It requires that a number of key elements be addressed (Figure 1): Landslide inventory, susceptibility assessment,

hazard assessment, risk assessment, management strategies and decision-making (Dai et al., 2002; Fell et al., 2005). Landslides present visible signs for reorganization, classification, and mapping in the field, completed by the interpretation of satellite imagery, aerial photography, or the topographic surface (Guzzetti et al., 2012). There are many methodologies for landslide hazard assessment using geospatial technologies (van Westen, 1993; Soeters & van Westen, 1996; Guzzetti, 2000; Dai et al., 2002; van Westen et al., 2006). The classification methods can be categorized as: (1). Landslide inventory methods (Soeters

and van Westen, 1996; Galli et al., 2008; Sumaryono et al., 2014). (2). Heuristic methods (Ruff and Czurda, 2008; van Westen et al., 2006; Safaei et al., 2010) (3). Statistical methods (Huabin et al., 2005) and (4). Deterministic methods (Hammond et al., 1992; Zhou et al., 2003). Landslide inventories are the simplest and the most straightforward initial form of mapping because they display the locations of recorded landslides and they are a significant factor of most susceptibility mapping techniques and hazard assessments for qualitative and statistical analysis (Wieczorek, 1983; Dai et al., 2002; van Westen et al., 2006).

They have a different purpose, which in addition to location also include information and data on the type of landslides, triggering factors (e.g., earthquake or intense rainfall) and information on landslide susceptibility (Galli et al., 2008). They therefore have different techniques for preparation, including landslide distribution analysis, landslide activity analysis and landslide density analysis (Soeters and van Westen, 1996).

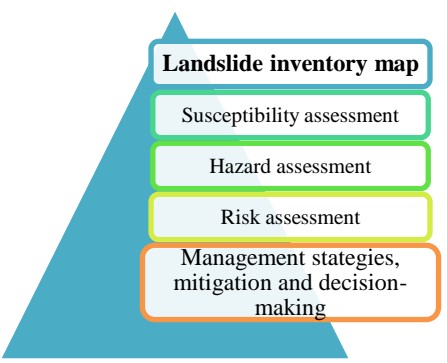

**Figure 1: Landslide inventory maps are the origin for landslide hazard and risk (Dai, et al., 2002; Fell, et al., 2005)**

## 2.1 Landslide data collection

Data collection includes desk and field studies and involve different activities ranging from low cost to expensive (Soeters and van Westen, 1996). The different techniques for data collection are divided into: 1. Image interpretation 2. Semi-automated classification 3. Automated classification and 4. Field navigation including total stations, GPS and recently GIS mobile. Field works are mostly carried out to classify groups of landslides triggered by an event, acquire data about characteristics of landslides, check inventory maps prepared by other methods, and improve visual interpretation of satellite images (van Westen et al., 2006; van Westen et al., 2008; Safaei et al., 2010). Landslide inventories can be characterized by scale and the type of mapping (Guzzetti et al., 2006) and they are developed by gathering historic information on different landslide events or Remote Sensing (RS) data (i.e. satellite imagery and aerial photographs) together with field verification using GPS (Soeters and van Westen, 1996). There are some examples of different methods using RS, LIDAR and comparisons of inventory maps (Galli et al., 2008; Pirasteh and Li , 2016). Landslide inventory data, hazard factors, and elements at risk (Figure 2) are the three main essential layers for landslide hazard and risk (van Westen, 2004). The landslide inventory is the most significant among them because it acquires the location information of landslide phenomena, types, volume, and damage (van Westen et al., 2008).

Historical landslide records and freely accessible databases have been developed for a few countries, (e.g. Italy (Guzzetti, 2000), Switzerland, France, Hong Kong (Ho, 2004), Canada and Colombia).  However, difficulties related to completeness in space and time are a drawback (van Westen et al., 2006).

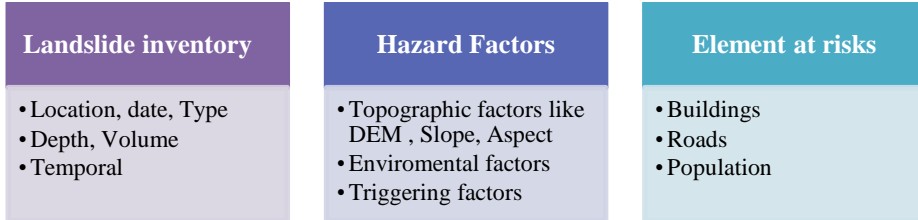

**Figure 2: Database for Landslide risk assessment and management (van Westen, 2004)**

## 2.2 Mobile and web GIS for landslide inventory

Many improvements in digital mapping and mobile GIS using Geospatial technologies have been revealed in the field of data acquisition for landslide hazard and risk which mostly are open-source. Following are examples of these technologies. The BGS digital field mapping system (BGS-SIGMA mobile 2013) includes customised ArcMap 10 and Ms Access 2007 which have customised two toolbars for mobile and desktop for digital geological mapping. The mobile toolbar was developed to capture data in the field on tablet PCs with integrated GPS units and the desktop toolbar focuses on data interrogation, data interpretation and the generation of finalized data. This is a free software however, it requires Arc Editor Licence (BGS, 2013). Geodata implemented a mobile application that can add hazards as point markers with an attached image (GeoData, 2015). Another prototype for landslide geomorphological mapping using Open-source Geospatial Foundation software such as MapServer and Postgis was implemented in the Olvera area, Spain to improve transportation and construction of roads (Mantovani et al., 2010). This application runs on desktop and focuses more on data management system and visualization of data. WbLSIS (Acharya et al., 2015) is a desktop conceptual framework for Web-GIS Based Landslide Susceptibility for Nepal with emphasis on data management. Another web-GIS tool was developed for landslide inventory using data driven SVG (Scalable Vector Graphics) and paper sketch maps (Latini and Köbben, 2005). Temblor is a mobile application for the purpose of visualizing hazard maps online anywhere (Temblor, 2016). Lastly, Global disk platform by UNEP is a Web-GIS platform which uses open-source to visualize hazard maps and other related data from many countries (UNEP, 2014) but data available in that platform is limited. There are few systems with an option of using mobile technology for landslide and hazard field surveys, while there are several related systems using satellite images and mobile GIS (e.g. a GIS mobile application (Bronder and Persson, 2013) for data collection of cadastre (cadaster) mapping using ESRI and Google Android SDK). Geoville has developed a highly-automated land-cover and land-use mapping solution that transforms satellite images into intelligent geo-information (Geoville, 2016). USHAHIDI can build tools to solve unlimited data acquisition, data management, mapping, and visualization challenges using multiple sources such as mobile applications, email, and twitter (USHAHIDI, 2015). All the above mentioned systems have some disadvantages for our study such as: limited access (BGS, 2013), limited drawing tools (GeoData, 2015) (e.g. point markers only), desktop GIS (Mantovani et al., 2010; Acharya et al., 2015), paper-field systems (Temblor, 2016), and limitations related to visualization and data acquisition (UNEP, 2014). There are different systems in mobile GIS and data collection; however, the possibility for having an open-source- mobile application, with an added satellite image in offline mode, precise mobile GPS, easy and fast drawing tools, advanced visualization, and database management system, for landslide data collection is quite necessary.

## 3. Implementation

The ROOMA application was developed to complement conventional remote sensing for landslide inventory creation. It is based on a prototype web and mobile GIS application including an online database to overcome some of the aforementioned problems related to landslide database development. This approach compensates the lack of landslide inventories and precise

topographic process, and decreases the resources and time needed for data storage and updating. In addition, the combination of the ROOMA data collection in the field with GPS and satellite image as source maps can significantly improve the accuracy and quality of input field data. The satellite image added to the application significantly eased the exploration of this area and assisted the visual interpretation process. Figure 3 demonstrates the workflow of this method. Image interpolation coupled with field surveys enables the development of a range of GIS based maps including information such as landslide distribution, hazard, and damage infrastructure and a more complete database of landslide data and their characteristics.

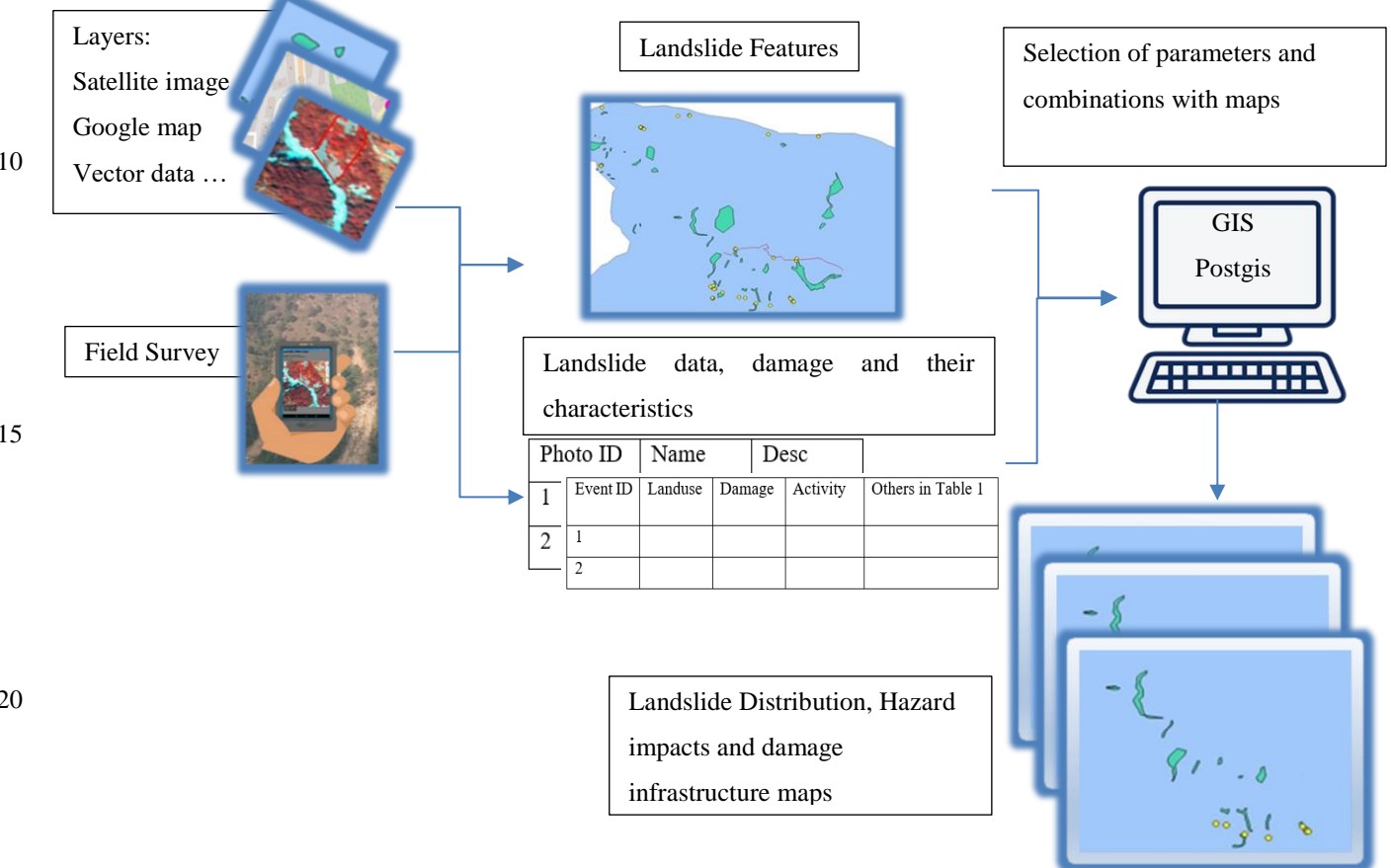

**Figure 3: Workflow of ROOMA where coupled image interpolation with field survey leads to asset of maps and complete database of landslide data and their characteristics. These different maps of landslide distribution, hazard, and damage infrastructure can be produced by manipulation in GIS.**

Landslides are created by and impacted by a large number of components, for example geology, land-cover, land-use practices and earthquakes. Table 1 illustrates different types of information which can be collected during field mapping of landslides using this application (Offline version). The first 3 rows in this table are compulsory to be filled in the field survey using mobile application (Landslide ID is given automatically); however, the rest of them can be completed later in the office if

needed. This will help the user to save time in the field by recording one specific characteristic of their needs than entering all characteristics while not needed in their work.

**Table 1. Landslide data and their characteristics in the ROOMA database: Landslide ID is given automatically and Landslide Name and Shape are obligatory fields**

| Seq. | Field Name | Description |
|------|-----------|-------------|
| 1 | Landslide ID | Numbers of landslides |
| 2 | Landslide Name | Name of landslide |
| 3 | Shape | Point, Line, Polygon |
| 4 | Date of event | 01-01-2015 |
| 5 | Date of record | 01-01-2015 |
| 6 | Type of material | Debris, Earth, Rock |
| 7 | Type of movement | Slide, Flow , Fall , Rotational slump, Flow slide |
| 8 | Land-use Features | Forest, Road, River, Agriculture field, House… |
| 9 | Damage | Road, House, School, Forest, Communication line… |
| 10 | Triggering factor | Rainfall, Earthquake, Human activity, others |
| 11 | Reactivated? | Yes, NO |
| 12 | Presently active? | Yes, NO |
| 13 | Possible reactivation? | Yes, NO |
| 14 | Hazard Degree | No hazard, Low, Medium, High |
| 15 | Possible Evolution | Up, Down, Widening |

Data on elements at risk in an affected area (houses, schools, inhabitants, road networks, utilities, etc.) form the basis for landslide risk assessments. Importance is commonly placed on data related to houses and people; though in this work, emphasis is given to buildings, road networks and infrastructure. Generally, data on elements at risk are collected by satellite images

and result in the production of versatile databases; however, for this prototype, elements at risk can be recorded directly in the field along with other attributes of landslide event data (Table 1). Elements at risk have different characteristics including spatial (the feature in relation to the landslide), non-spatial (e.g. temporal data such as inhabitants) and thematic characteristics (e.g. material type of the buildings). Saving land-use features (elements at risk which are damaged or not) along with event

data (e.g. hazard and damage to infrastructure) in the field is another advantage of the ROOMA application compared to abovementioned systems.

Figure 4 demonstrates different types of spatial and non-spatial data that are recorded in the ROOMA data model. Each table represents name and type (e.g. integer) of the column. The only mandatory (Marked as nn: Not Null) data to be recorded are the features and name of event, the remaining data can remain null and be filled in later if necessary. Upon the creation of a new "studyarea" table in the online platform, a new database and schema are created dynamically to store all events related to that "studyarea". Each "studyarea" has many "event" tables which can record information on landslides and the view-points (as Geometry POINT) where this event is mapped. Each event is associated with different feature tables (feature_polygon, feature_line or feature_point table) and "photo" tables that represent landslides, damage (elements at risk), and photos. The data in these tables are automatically created from GeoJSON-text files which have been uploaded to the ROOMA online version. This data model made it easy to query on and analyze data based on each "studyarea". The case study area for this project is explained in section 5.

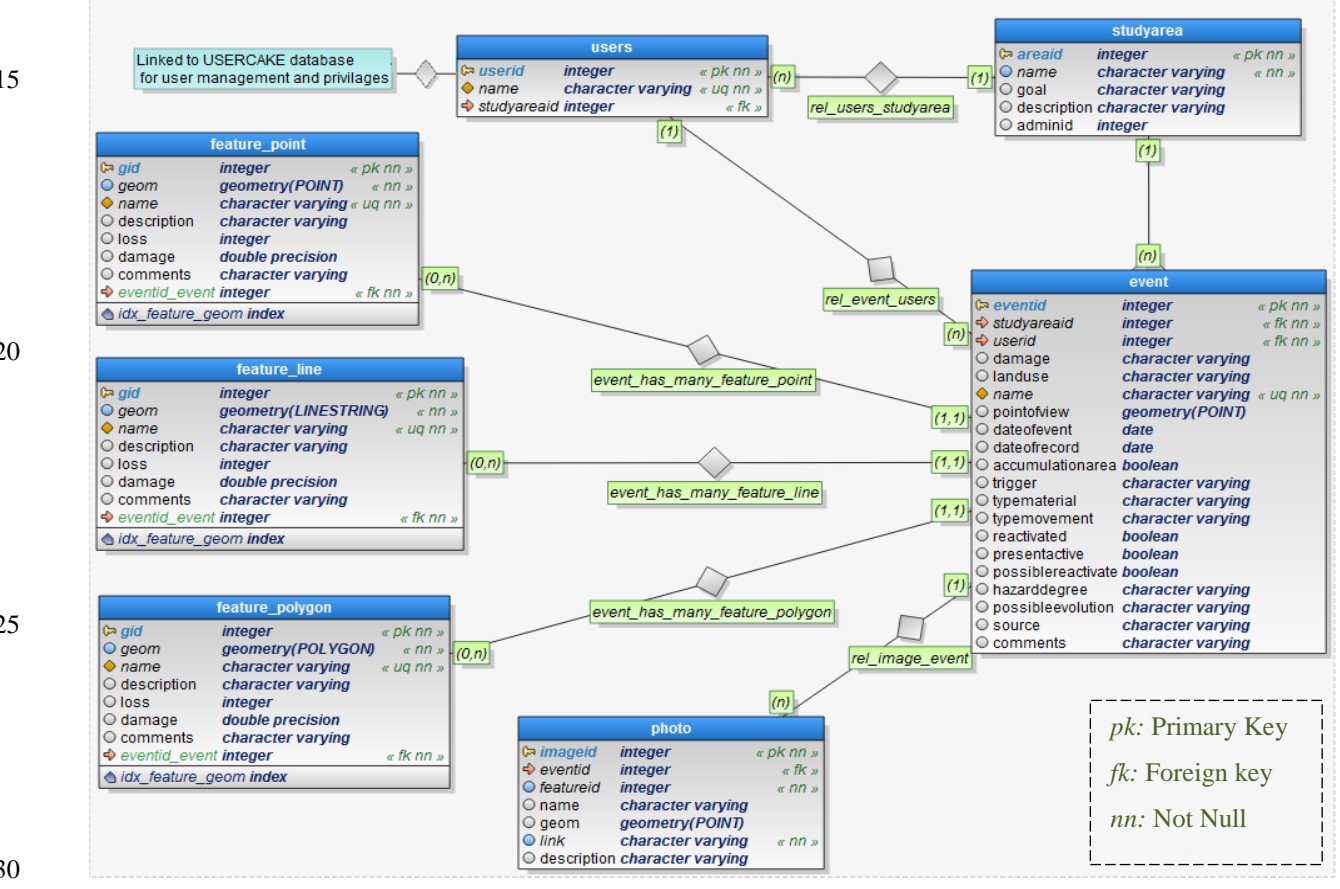

**Figure 4: Data model of ROOMA: Database is automatically created from GeoJSON-text files which have been uploaded into online version of ROOMA.**

### 4. Technology and Platform: Mobile GIS

Free and Open-source Software for Geoinformatics (FOSS4G) have significantly improved the efficient mapping and management of post disaster and impacted areas around the world (UNEP, 2014; USHAHIDI, 2015; Geoville, 2016). GIS can integrate different layers of spatial data on landslide occurrence to define the effects of various parameters.

There are new developments in Open-source Geospatial technology for visualization and analysis landslides, including: (1). Digital acquisition and editing tools (Leaflet, 2015), (2). Advanced geo-visualization (BoundlessSpatial, 2016), (3). Enhanced integration with satellite imagery using TileMill (Mapbox, 2016), (4). Combination with database management systems (PostgreSQL, 2015; PostGIS, 2015; MySQL, 2015; UserCake, 2015) and (5). Amplification of the accuracy by using mobile GPS (Cordova, 2015).

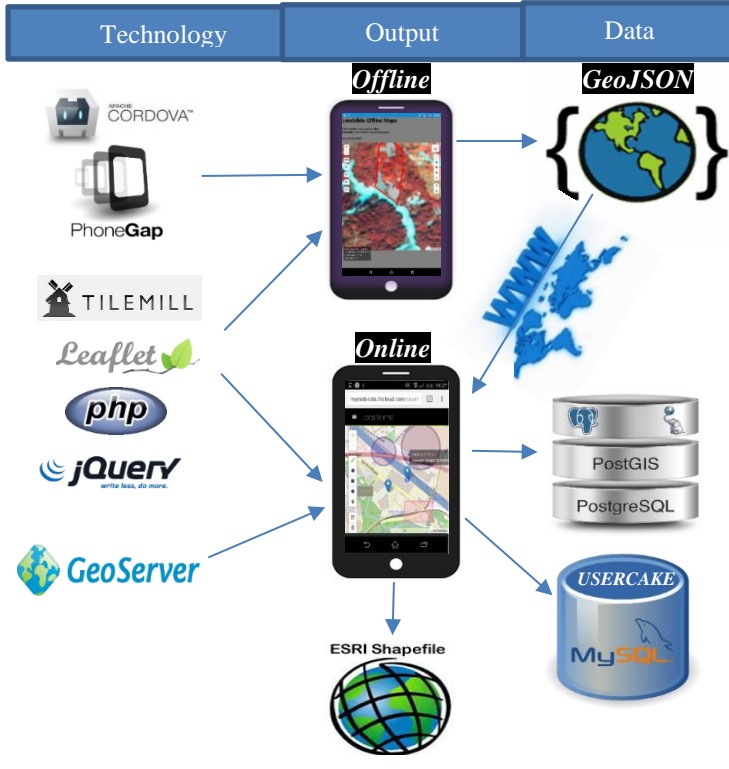

**Figure 5: Technology (Cordova and PhoneGap) used by ROOMA, upon which the offline version is built. The online version is based on tree-tier architecture which includes the presentation, application and data layers. The presentation layer is based on Leaflet, jQuery, and JavaScript. Application layer uses PHP to connect to GeoServer and database. The data layer is composed of both MySQL (UserCake) and PostgreSQL (Postgis).**

The offline Android component of ROOMA is implemented using Cordova (Cordova, 2015) and PhoneGap (PhoneGap, 2015) (Android environment based on JavaScript) to simplify data collection in the field in remote areas where internet access is poor. The satellite images are transferred to Tiles using TILEMILL (Mapbox, 2016) and added to Leaflet map library in both

online and offline version. The online version of this application is based on client–server software architecture pattern, (tree-tier architecture) which includes presentation, application and data layers, developed and maintained independently (Williams and Lane, 2004). Both offline and online versions use client-side jQuery and leaflet libraries. The different geometrical features (points, lines, and polygons) for landslide data by different descriptive attributes e.g. type, date, activity, triggering factor and

hazard degree are given in GIS format called GeoJSON (GeoJSON is a format for encoding a variety of geographic data structures which is similar to Keyhole Markup Language (KML) format, GeoJSON, 2015) using Leaflet map. The data can be exported to GeoJSON-text files and uploaded through the internet to the online component where the main database is located. This enables the collection of data from multiple data collectors into the same database. Server-side is based on PHP, which transfers data to the database and saves the output of Leaflet map in GeoJSON. The geodatabase was designed to incorporate

geospatial data acquired in the field, delivered as an input to the system (e.g., type, shape, volume, date, triggering factor, hazard degree) in relation with elements at risk data (e.g., building information, road network, damage information) connected to a specific event (Figure 4). The FOSS4G technologies selected for this module were PostgreSQL 9.4 (PostgreSQL, 2015) and Postgis 2.1 (PostGIS, 2015) for spatial database management. The GeoServer 2.6(Geoserver, 2015) module, in connection with Geodatabase (Postgis), is delivered for visualization and spatial analysis. This component brings a complete and up-to-

date description of the different layers including a landslide event layer, elements at risk layer and detailed information of landslides including event descriptions and photo mapping if any georeferenced photos are uploaded to the online version. Finally, the outcomes are captured and shown through GeoServer and OGC services such as Web Map Service (WMS) and Web Feature Service (WFS) as well as being exported as shapefile format and visualized in other GIS software. UserCake library (UserCake, 2015) is an open-source library in PHP which using MySQL database (MySQL, 2015) to improve the user

management and authentication. Two type of users are available in this system: Public and Administrator. Based on their privileges, they can access to different components of the online version. For example, only the administrator can define a new "studyarea" and assign that to different users. Figure 5 displays the technologies and the frameworks of this prototype.

The offline component of ROOMA (Figure 6) contains the following modules: (1). Geolocation using GPS on mobile or tablet, (2). Map with combination of multi-source base layer (OpenStreetMap, Satellite image, vector data can be seen in figure 8)

(3). Map drawer (Line, Polygon, Rectangle and Marker) (4). Satellite image as the base layer and (5). Saving options as GeoJSON-text file in the offline mode. The mapping process is quick and easy: various types of satellite images are used as base layers for easy identification of objects on the map (Figure8: b), upon which different features can be drawn on a map drawer after geolocation. The online component presents more modules in addition to the map and geolocation options (Figure 7 and 8): (1).Saving online events directly to database, (2). Photo mapping, (3). Photo and event clustering, (4). User privileges

(5). Data storage and analysis, (6). Import from/Export to Shape files.

The user can save or upload these features as one event and define additional characteristics as mentioned in table 1. Figure 7 and 8 illustrate how an administrator can view different landslide events in the online version with the possibility of clustering events (Figure7), different base layers (Figure 8: b), and editing events (Figure 8: a) directly into the online database.

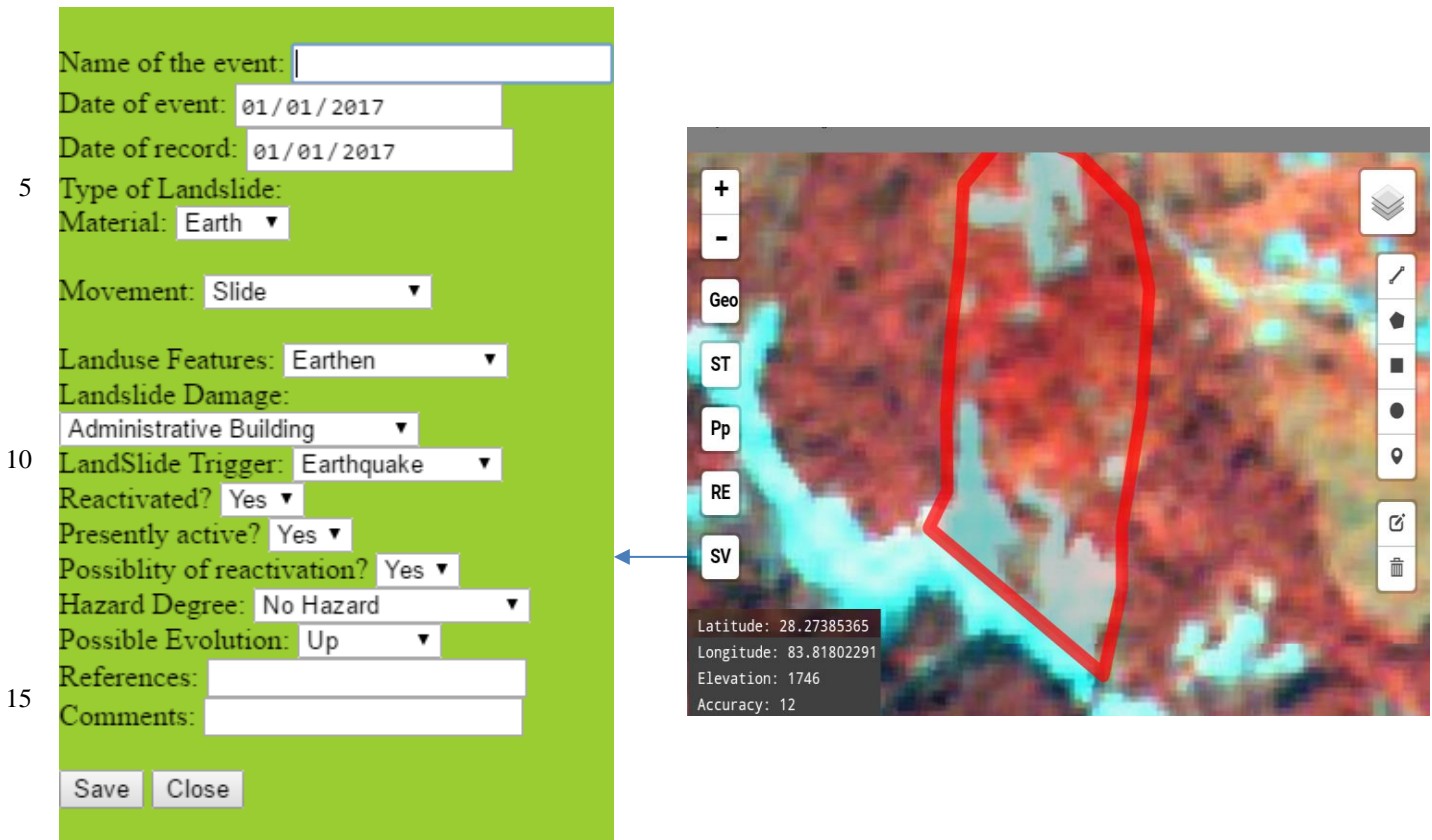

**Figure 6: Offline Component with a satellite image as a background: Geolocation (Geo), Stop Geolocation (ST), Show all the attributes in a pop up window (Pp), Reset the map (RE), and Save as GeoJSON-text (SV) by filling the green from.**

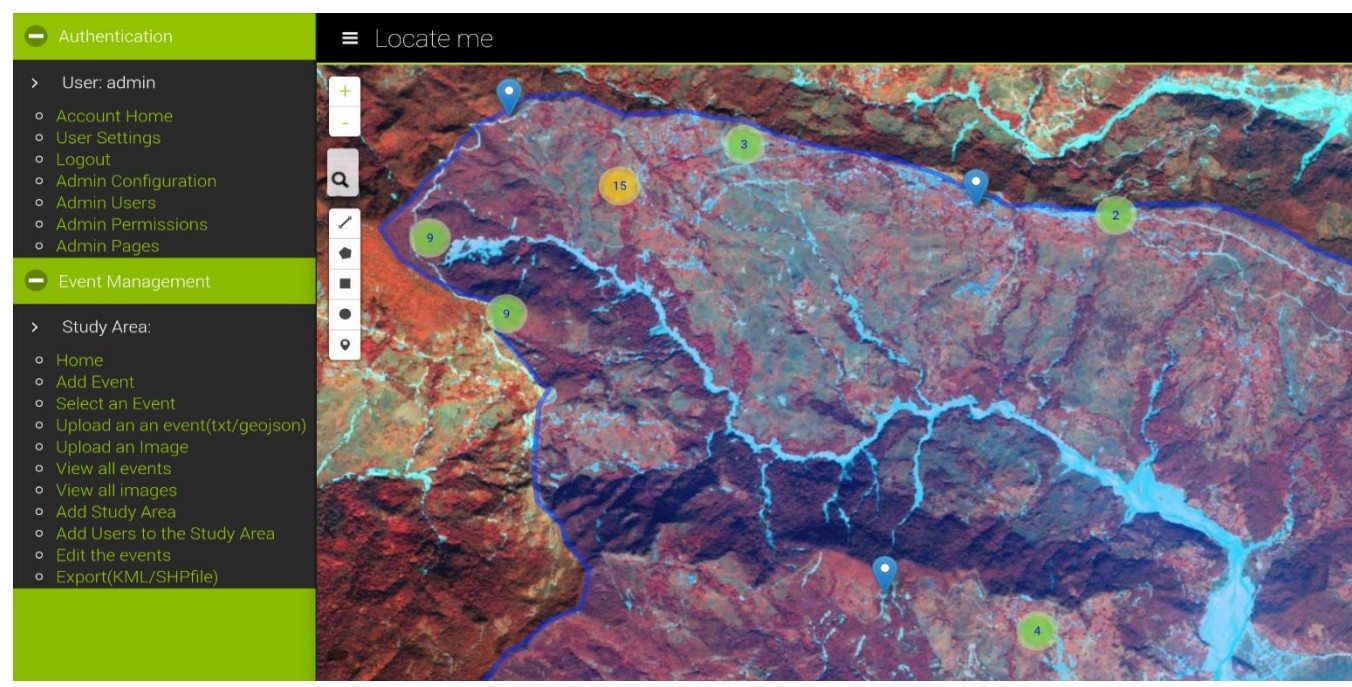

**Figure 7: Online component: User authentication and event management as an admin user: all the recorded events shown as cluster points**

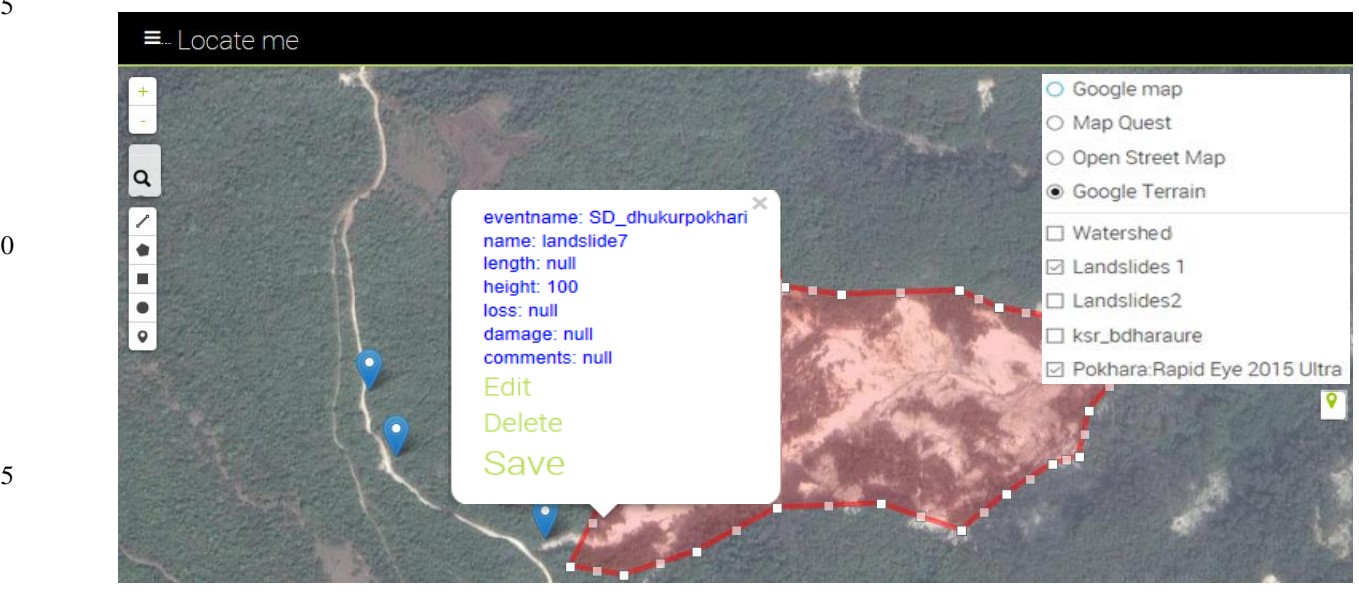

**Figure 8: Online component: A landslide event with the options of editing the feature directly into the online database and adding different layers as base layer such as google map, a shapefile or satellite images (Pokhara: Rapid Eye 2015).**

## 5. Case Study

Many landslide studies have been conducted in the Everest regions (Gupta and Saha, 2009; Bajracharya and Bajracharya, 2010; ICIMOD, 2016; Sato and Une, 2016). The 7.6 magnitude earthquake in Nepal on 25th April 2015 and a series of aftershocks significantly increased the risks of landslides (Collins and Jibson, 2015). Nepal has a high natural geological fragility further increased by the 2015 earthquake, which triggered several thousand landslides (Collins and Jibson, 2015; ICIMOD, 2016). The ROOMA application was tested in the Phewa Lake Watershed (123 $km^2$) in Western Nepal, Kaski District (Figure 9) where authors have been monitoring landslides since 2013. An intense rainfall event (315 mm in 4 hours) killed 9 people on 29 July 2015 in Bhadaure-5 near Pokhara and another 25 people were killed nearby Lumle in Parbat District (BBC, 2015). It was very hard to identify all landslides and their properties through image interpretation, so the impetus for field mapping was very high. The ROOMA application was field tested for a rapid assessment of landslides triggered by this event or reactivated along with their land-use characteristics and damage to houses, schools, roads, rivers, agriculture fields and forest area (Figure 10).

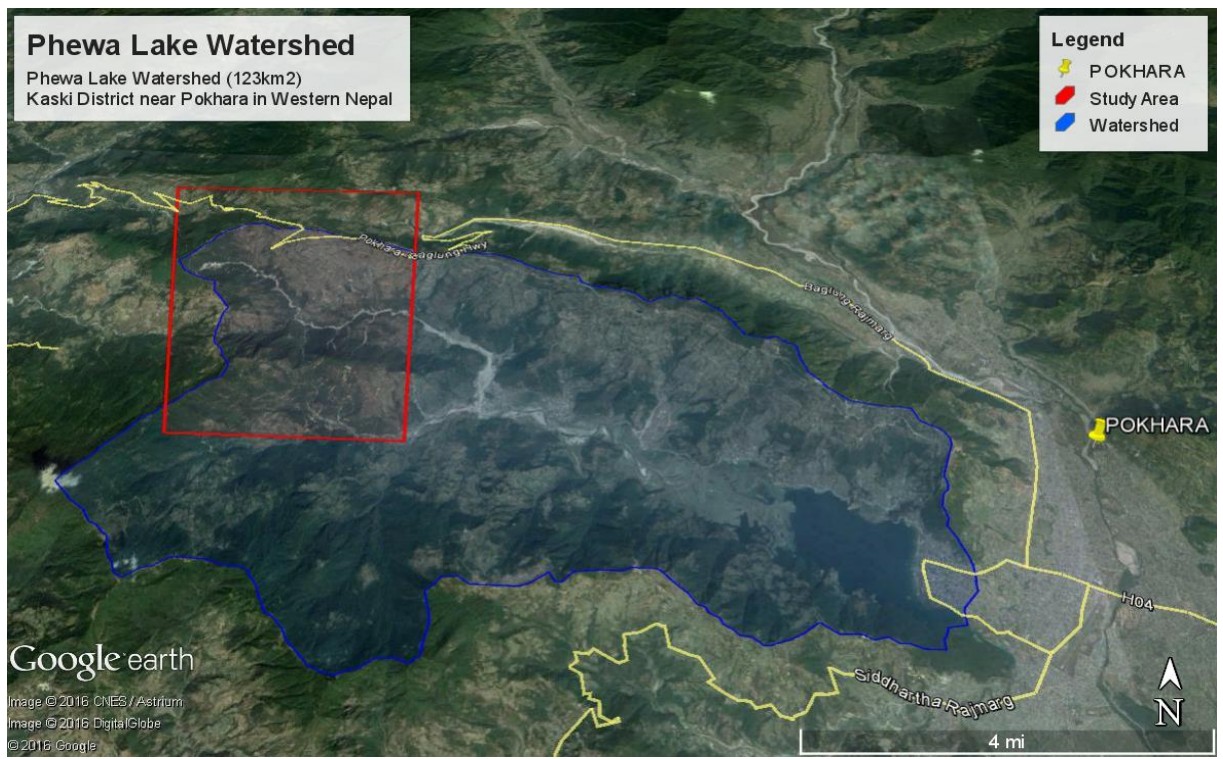

**Figure 9: Google earth image for Phewa Lake watershed, Pokhara, Nepal**

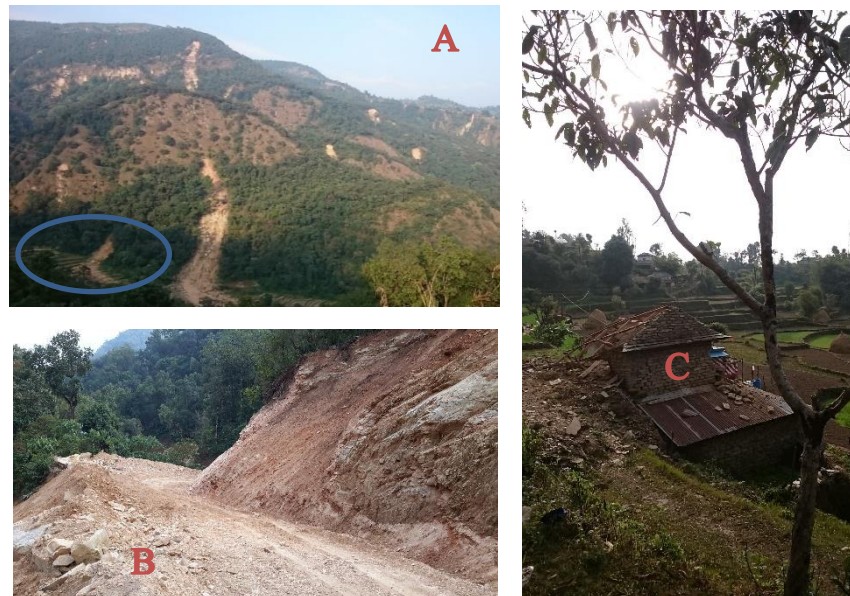

**Figure 10: A: Photo of the area with several landslides near Phewa Lake watershed in Nepal and the damage to agriculture in blue circle, B: Photo of one landslide and damage to the road, C: Photo of a house damaged by a landslide in the ame area**

## 6. Results

Two days of field work were conducted in the Phewa Lake watershed, using the ROOMA application, which used medium resolution satellite image (GeoEye 2015, 5-meter resolution) to map 59 landslides. The mapping of landslides (using polygons) was accompanied by data collection on land-use features for each event (e.g. roads, rivers and forests) to give better indications of surrounding features. Mobile-GIS using satellite images and offline version gives an opportunity to see landslides that already existed and their distribution in that area.    The data were collected in the field using the offline version of this platform, either close to road or from a distance. This enabled easy interpretation of landslides which would have been difficult to access otherwise (Figure 11 and 12). Figure 11 represents a new landslide documented near the road that was not visible in satellite image and figure 12 shows a larger landslide which was located within a distance and clearly visible in image interpretation. Most of large landslides were mapped by distance. Figure 13 shows the distribution of landslides in an area where most landslides occurred in the center of the Phewa Lake watershed.

All data were uploaded to the online version and then exported to a shape file, while preparing the maps (Figure 15) were performed in QGIS2.6.1 (QGIS, 2015).  Data obtained from the field survey were successfully analyzed in the open-source GIS such as distribution of landslide type, material, elevation, damages, surface areas, and volume. In this article, we present some selected results. For example, all the information about land-use characteristics and their damages for different landslide

were gathered individually in our database and can be useful for more detailed analysis. Graph in figure 14 represents that a majority of the landslides occurred near forest areas and most damaged areas were related to forest, roads and agriculture.

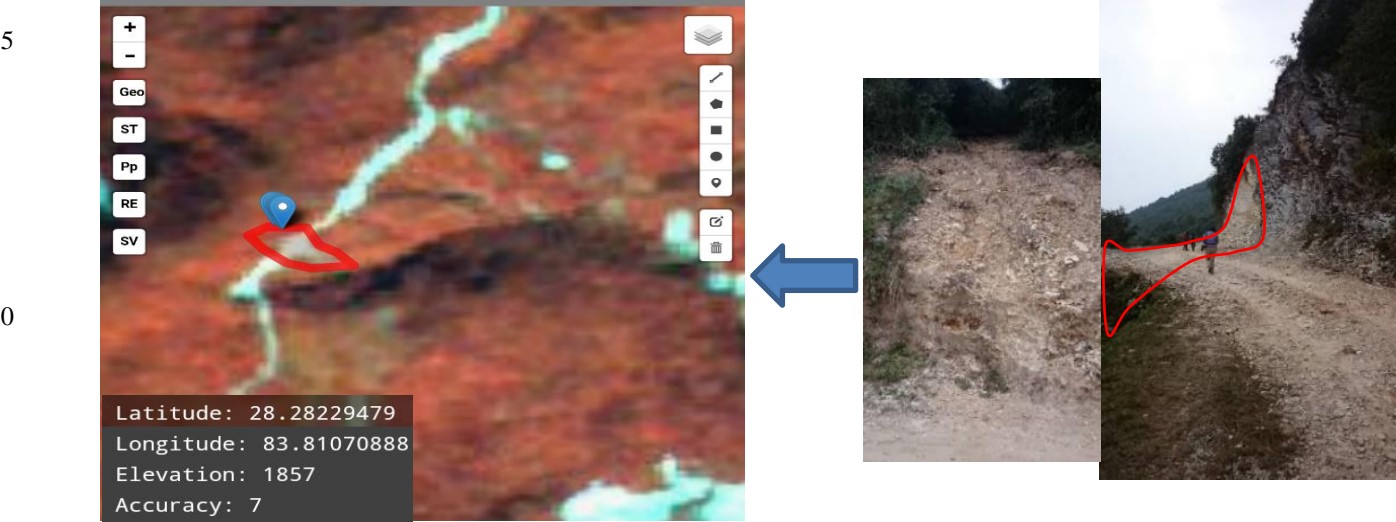

**Figure 11: Data collection close to the event where usually a landslide happened near a road and was possible to access**

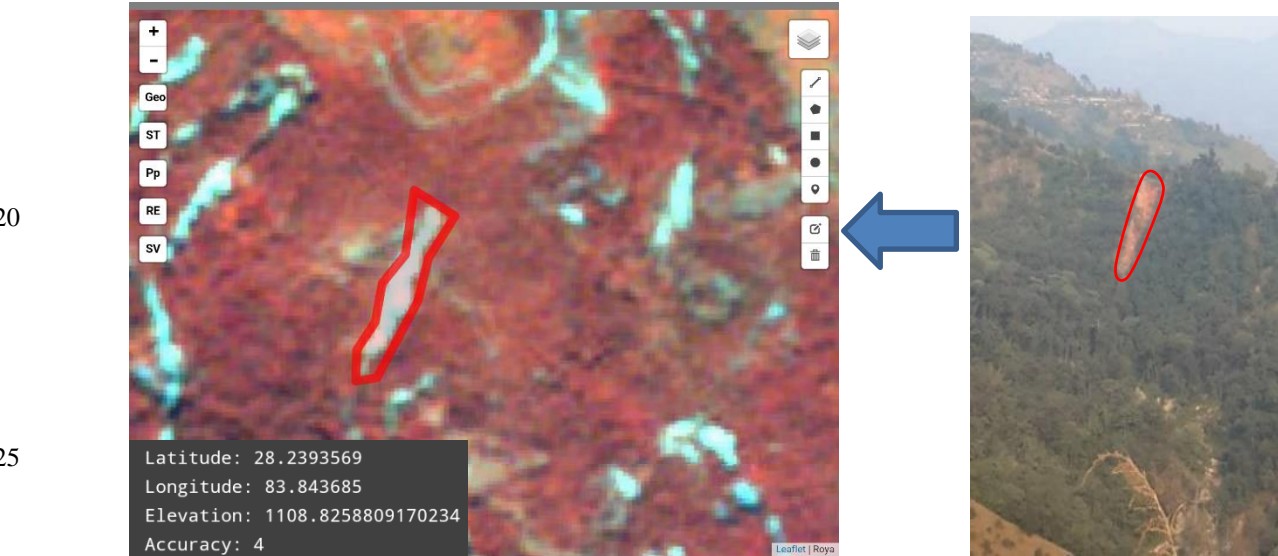

**Figure 12: Data collection by distance where was difficult to access however was easy to locate in the map using geolocation and satellite image**

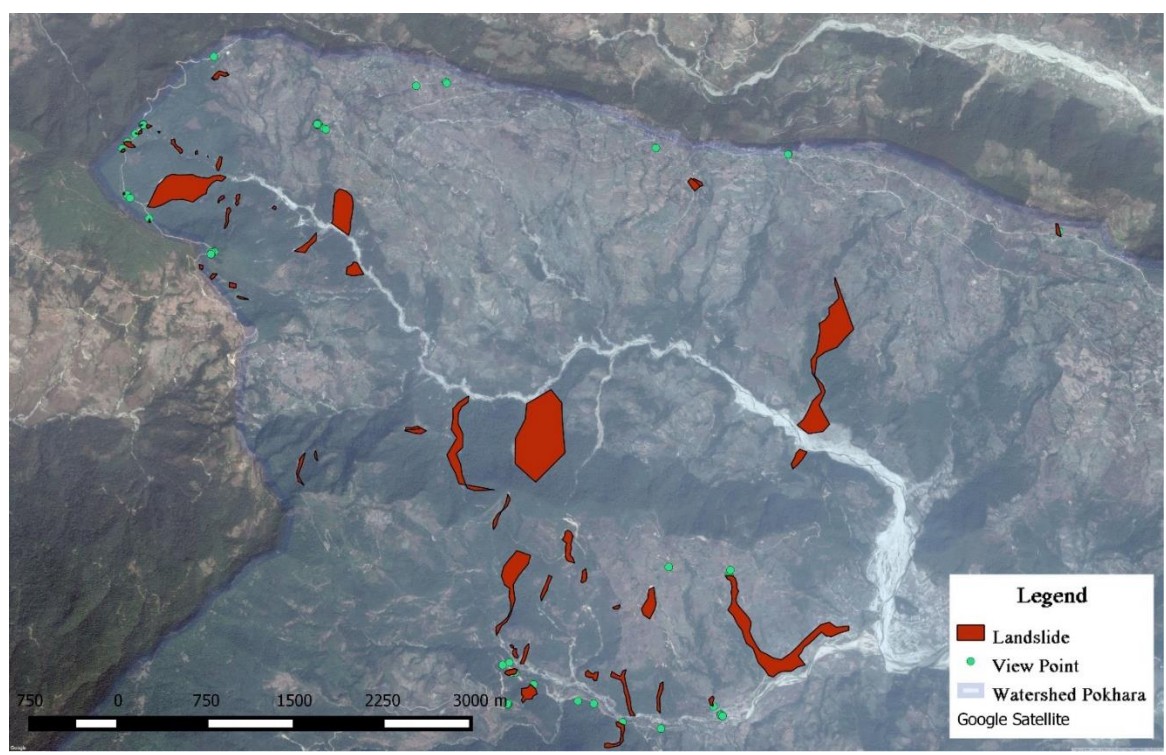

**Figure 13: Distribution of landslides in Phewa Lake watershed based on the two-day data collection**

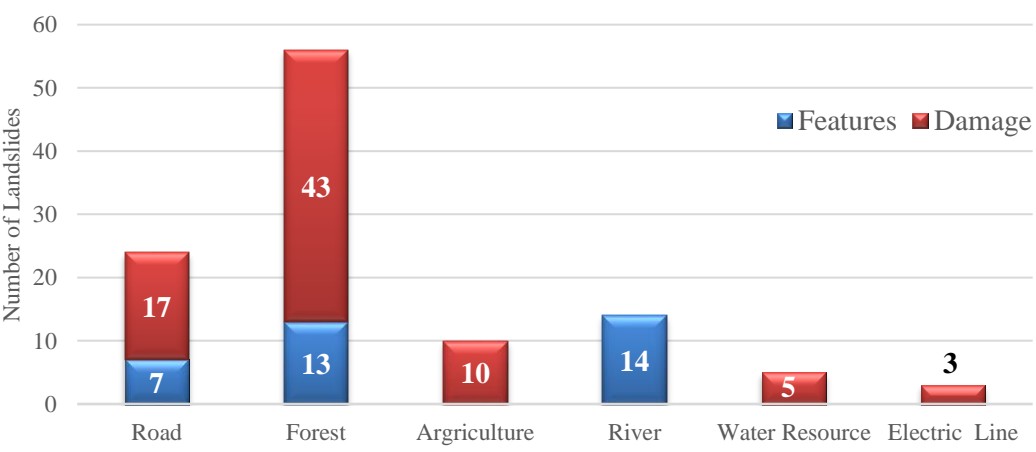

5    **Figure 14:  Relationship between features and landslides damage: for example 56 landslides occurred in forest and of these, 43 damaged the forest (Red = Damage).**

Moreover, further analysis of land use/cover changes has been carried out based on visual interpolation on a multispectral satellite image (SPOT 2016, 2 meter resolution) acquired in 2016 after this field checking. This image improved the quality of the polygons, nevertheless landslides are more difficult to identify as vegetation grows quickly. Principally, this ground truthing brought the confidence for further mapping (177 Landslides mapped afterward) of the additional smaller landslides that were not mapped during the field survey. Figure 15 shows these landslides on the map.

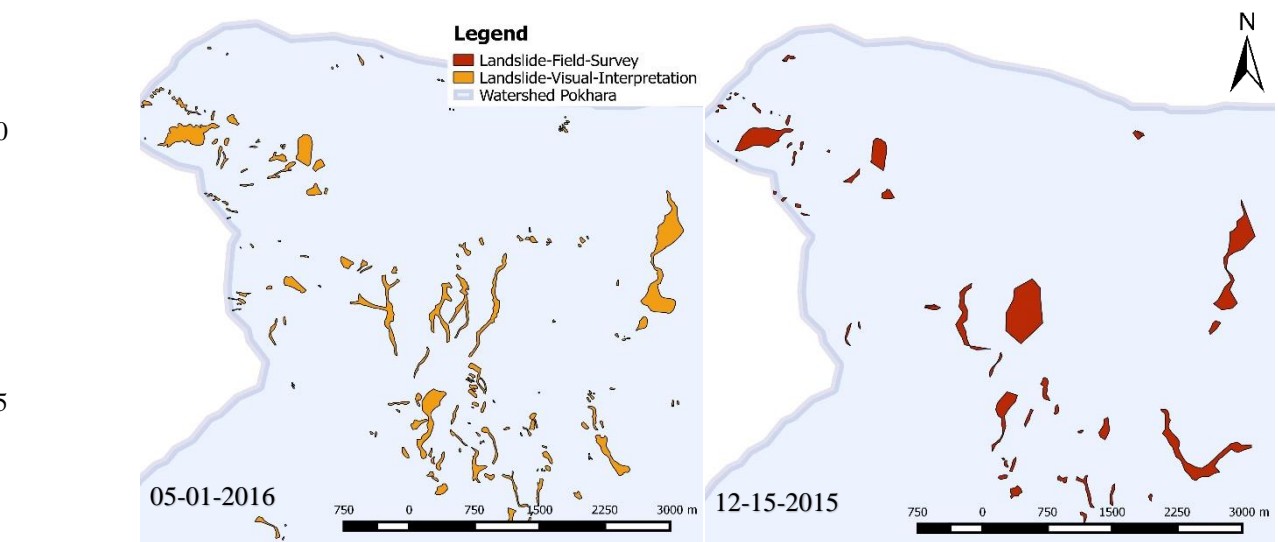

**Figure 15: Maps of landslides by using field survey (red polygons) and visual interpretation (orange polygons)**

The advantage of a mobile version with field survey compared to a mapping using only GIS and high resolution satellite images (in office) is that some feature characteristics of landslides are not visible on satellite images; therefore, coupling satellite image interpretation with field observation allows one to identify better the type of landslide even using a medium resolution satellite image (~5 m). Figure 16 shows such an example that the detail mapping on standard GIS permits to identify active landslides in the gullies, i.e. debris-flow and shallow landslides, while the lower resolution image coupled with field survey permits to identify larger landslide. Landslides linked with the gullies are often at the limit of the larger one, indicating landslide activity.

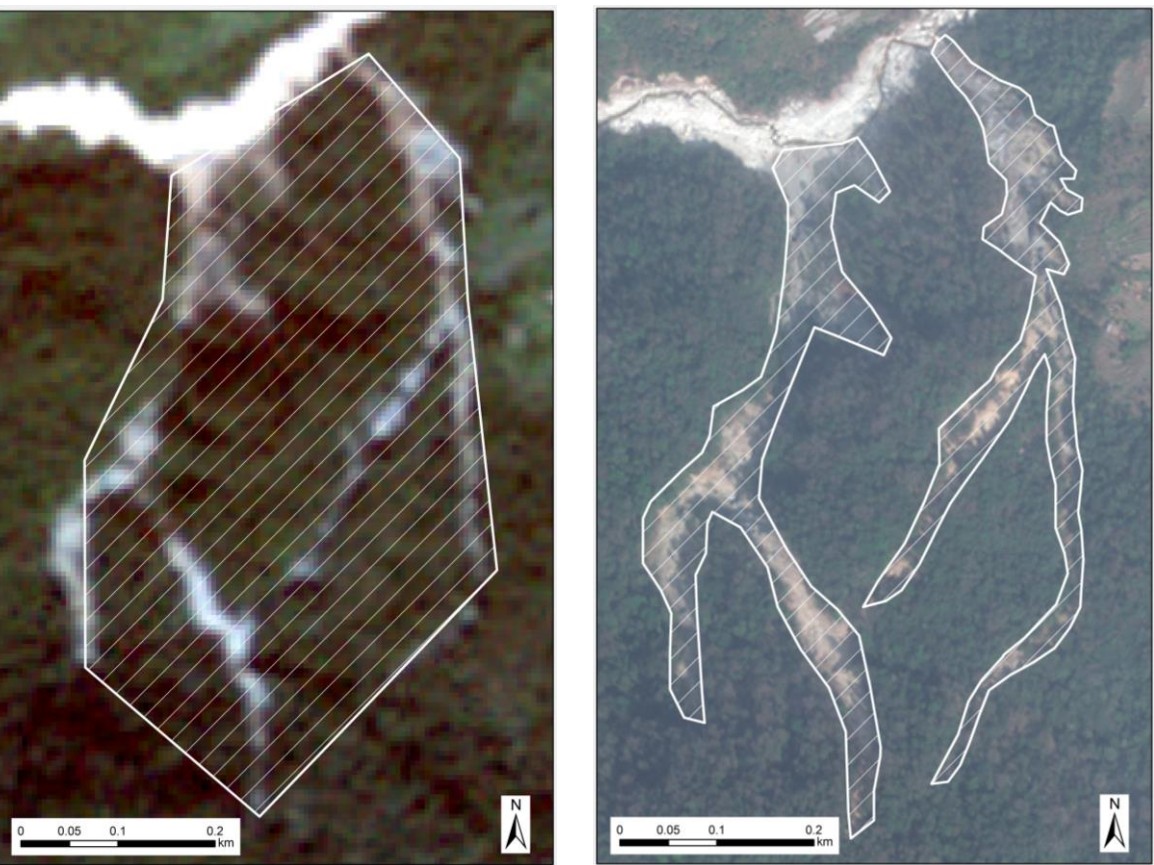

**Figure 16: The map on the left shows the lower resolution image coupled with field survey and the map on right shows the same area with the detail mapping on standard GIS. Field survey helped to understand that this is one larger landslide which is covered by vegetation however office work shows it as two separate landslides and ignored the part of landslide covered vegetation.**

## 7. Concluding remarks and discussion

Landslide inventories define vulnerability, hazard, landslide susceptibility and risk by investigating information on type, patterns, distribution and slope failures (Guzzetti et al., 2012). Earlier publications on landslide hazards shows that considerable developments have been accomplished in the last decade: GIS tools are now crucial for landslide assessments, however, the generation of landslide inventory maps (LIMs) including elements at risk and larger scale online databases have been developed but may be out of reach for data poor countries. The development of an offline rapid mapping application can provide a significant technological leap and save valuable resources. The value of landslide inventories relies on the accuracy and certainty of the information which is problematic to define (discussed in introduction) however, different mapping approaches on open-source geospatial technologies, can significantly simplify the production of these maps. Furthermore, the ability to use the open-source software indicates that analyses can be carried out without incurring the high costs associated with software acquisition, a particular advantage for developing countries, researchers and government officials.

Results for this paper are: (1). Android application (2). Testing the application (3). Analysis and comparison with similar work. This application incorporates rapid, economic and participatory methods for mapping landslides. It uses satellite images as multi-source map and enables multiple data collection to finally be collated in a centralized database. Data can be acquired in an offline version using an Android device or an online mode using all browsers in PCs, tablets and mobiles. The study was applied for mapping landslides in post-earthquake Nepal, but it can be applied for other hazard events such as floods, avalanches, etc. The result has been compared to the same study conducted remotely using image interpolation, and it shows that coupled field mapping with satellite image can improve the quality of landslide hazard and risk mapping.

Considering all the difficulties stated in this work (mentioned in introduction) for example difficulty to access the landslide and damage area, we did not face any specific issues during testing this application. Mapping a landslide is typically carried out based on the experience of the expert however, through mobile GIS, this application is easy to be run by non-experts and the general public. A combination of satellite data and web-GIS technologies provides an ideal solution for landslide hazard and risk data acquisition especially when more high resolution satellite images are freely available. The paper concludes that the ROOMA tool aims to increase the quality and speed of LIMs which can improve the quality for susceptibility, hazard, risk assessments, and landscape modelling.

The system is being further field tested for a future improved version; thus, this offline version can be improved by adding more components for distance calculation, continuous lines sketching, recording foot paths and merging the GPS located camera with the azimuth of data to help generate 3D models of the area.

This study can be enhanced through several of new developments to ROOMA, e.g. adding topographic data such as DEM and spatial-temporal modelling in order to increase accuracy. More effort is needed to incorporate and define vulnerability components, in order to generate risk maps. Finally, it is essential to integrate a spatial decision support system to use such data for landslide hazard and risk assessments for both stakeholders and local authorities.

**Acknowledgement**

We would like to thank Faculty of Geoscience at University of Lausanne and EPIC team (Ecosystems Protecting Infrastructure and Communities) for the funding of this project. We appreciate Institute of Engineering, Department of Civil Engineering at Tribhuvan University in Kathmandu, Nepal for their supports, friendship and leadership, and likewise their efforts for finishing this project. Finally, we would like to thank Professor Cees van Westen at ITC in Netherlands and Professor Brian G McAdoo at Yale-NUS College in Singapore for testing the application and for their helpful feedbacks and comments, some which are mentioned above.

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
