# Peer review of "An Offline-Online WebGIS Android Application for Fast Data Acquisition of Landslide Hazard and Risk"

_Natural Hazards and Earth System Sciences, 2016_

## Referee Comment (RC1) · Anonymous Referee #1 · 13 Sep 2016

Dear Author, all details of review are within PDF file. I list specific comments within it. Personally I consider the scientific aim interesting with a quite good approach. My comment within report is a major review and I am glad for next step of manuscript. Regards

Please also note the supplement to this comment:
http://www.nat-hazards-earth-syst-sci-discuss.net/nhess-2016-267/nhess-2016-267-RC1-supplement.pdf

**Supplement:**

Dear Author,

I was glad to review your paper, interesting for contents and final aims. I list hereafter tasks you should review for publication. I will reconsider your paper after major revisions. The argument is encouraging, but actually incomplete for methodology and output. Specific reviews of chapters appears hereafter.

- Paper is written with current and regular languages, you adopt technical jargons but not with accurate quality of single chapters. Introduction and background list a context in which the paper reveals the output. Concerning landslides, you introduce susceptibility, hazard, risk, survey types, patterns, distributions, statistics of slope failure, management strategies, risk assessment… Monitoring systems miss in the list, to complete landslide treatment from geomorphological and geophysical point of view (too much…) The general description is quite systematic but the references are not complete, because topics are vast, complex and only cited, often with a general redundancy. The database chapter explains technologies for data collection and GIS system for inventory of landslides in a territorial context. The methodology chapter should be the core, but it lists data and characteristics required for single landslide, integrating characteristic of landslide with element at risk, unclear and indefinite. Technology and Platform appears as core of the output with a copious integration of know-hows and available solutions. The study area illustrates landslides with subsequent results, but without specific enlightenment. Susceptibility map with spatial modelling and many types of data emerge again within chapter, confusing the real output.
- The paper contains an innovative issue but not well ordered. ROOMA has a complex architecture, for gathering and field survey. The activity includes android environment to deploy free and open solution for data collection. The idea is offering a crowd system, to combine user-friendly tools for geospatial activity on field. The participation could include contribution in a large area, with rapid methods for slide mapping. This challenge is innovative, but not included as priority in the paper, because not clearly linked to the landslide dataset required. The paper overbalances interest on all details of risk assessment, deleting request of landslide data and technology adopted.
- Digital field survey exists since around 10 years, within geomorphology final aims, controlled by high precision in GPS location and field GIS integration (MapIt, ArcPad, Geopaparazzi, GISTrimble, and other FOSS4G solutions). You mentioned tablet and mobiles, but the advantage of platform is android environment, customizable and free of costs. The advantage is the online-offline, independent by bandwith, offering a tool definitely fast, user-friendly and low-cost. These advantages are not enlightened. Clear problems could be bug-fixing, GPS precision. The advantage of online-offline includes clear benefits, you need to highlight them compared to traditional field survey.
- You explain the aims as android mobile application on both Offline-Online access. The aim is a fast and storing of data. The visualization and drawing tool is based on central database available to services (mobile, PCs and web browser). Data management improves in hazard event mapping as you declared, but the aim is too general, not simply split.

- Mobile-GIS has a clear gain, but limits due to dimension of mobiles, resolutions, spatial tools available, zooming, spatial extent, route, snapping and editing tools have to be revealed. If compared to desktop GIS you have to explain the difference. ROOMA has a mobile solution, introduced in data transfer. Specify the content of slides and clarify what users can do on the field. Mobile-GIS with GPS are tools to increase efficiency in data collection.
- Online-offline is an interesting approach with Geojson. How can data be saved and furthermore included in geospatial analysis? Explain a bit better the technology adopted. The architecture is not well shown.

**1. Introduction**

Landslides incorporate all types of mass movements on slopes (Varnes , 1984) and can be triggered by various external events such as intense rainfall, earthquakes, water-level changes, storm waves or human activities. The location, the time of event and the types of displacement can be recorded in a landslide inventory map. In this paper, we do not distinguish between "landslide map", "landslide inventory map", and "landslide inventory". Landslide maps are important factors for landslide hazard and risk assessments, particularly if there is a significant number of landslides with different types, dates, volumes and trigging factors (Coe , et al., 2004). They can be produced using diverse methods however the selection of techniques relies on the size of the area, the resolution , the scale of the map, land use, land cover, soil and geomorphology (Coe , et al., 2004; Guzzetti , et al., 2006; Hungr , et al., 2014). Formulating and documenting landslide maps is essential to define landslide susceptibility, hazard and risk and to survey types, patterns, distributions, and statistics of slope failures. However developing complete landslide inventories are difficult, due to accessibility, the dynamic nature of landslides and also the time required (van Westen, et al., 2006). Conventional techniques lead to the development of landslide inventories mainly based on the visual interpretation of satellite images, assisted by field surveys. Typical issues for creating these maps include (Guzzetti, et al., 2012; van Westen, et al., 2006; Safaei, et al., 2010):

1. All methods for developing landslide inventories have long process and intensive resource.

2. Landslides are often small with high frequency of occurrence which located in remote areas and difficult to access

3. Landslides often have different characteristics which require them to be mapped and documented individually.

4. The lack of landslide documentation and databases are the main disadvantages in the evaluation of landslide hazard risk.

**Commentato [S1]:** Please simplify differences, is not clear for data typology and requirements.

**Commentato [S2]:** The access of field area is difficult (dynamic nature, what is?), because you require landslide inventory for final map.

**Commentato [S3]:** Long process and intensive resource. What are these parameters?

5. Limited damage data are available for landslides, which is why developing landslide vulnerability assessments is challenging.

6. The source of landslide inventories such as aerial photography, satellite imagery, InSAR (Interferometric Synthetic Aperture Radar) and LiDAR (Light Detection and Ranging) are expensive.

GIS for landslide susceptibility and hazards with respect to the type of data available, landslide type and potential extension have been described by several authors (van Westen, 1993; Guzzetti, 2000; Van Den Eeckhaut, et al., 2009; Carrara, et al., 1991; Dhakal, et al., 2000) . While the above authors have noted the importance of enhanced mapping, mobile-GIS offers technology for more effective ground-truthing and a rapid tool which can systematically fill a database, especially for unexperienced mappers. Currently, there is a high possibility to apply mobile-GIS including GPS and mapping tools to significantly increase data collection efficiencies.

In this paper, an offline-online application based on Geospatial Open-Source technologies (Called ROOMA : Rapid Offline-Online Mapping Application) is described to collect data on landslide events, hazard impacts and damaged infrastructure, which can be made readily accessible to authorities, stakeholders and the general public. This prototype provides a solution for preparing landslide hazard maps in relation with vulnerability. Besides, the advantage of an offline technology helps to map the events, especially in rural areas where internet is not available. This prototype has following objectives:

1. An android mobile application with possibility of both Offline-Online access

2. Fast and easy acquiring and storing of data and information

3. Advanced visualization and drawing tool 5

4. Central database with availability by different services (mobile, PCs (Personal Computers) and standard web browser)

5. Data management improvement in hazard event mapping and storage

The paper is structured as follows. In section 2, we first present the background, principles of the different approaches for landslide inventory, and the importance of landslide inventories maps in hazard and risk assessment. We also review some GIS tools that simplify field navigation. Then, Section 3 discusses the description of mapping method, with field survey for 10 preparation of landslide maps in relation with elements at risks. Section 4 illustrates the architecture and platform using open source geospatial technologies to map landslides by using an android application. Section 5 and 6 focus on study area and results. Finally, section 7 concludes by discussing the advantages of mobile-GIS, with the future outlook of producing landslide hazard and risk.

**2. Background**
* * *
**Commentato [S4]:** "GIS for landslide susceptibility and hazards" is redundant in the paper. These are complex and fundamental steps in risk assessment, with methodology since 10 years… Why do you consider these measurements as concept linked to landslide inventory?

**Commentato [S5]:** "mobile-GIS offers technology for more effective ground-truthing and a rapid tool which can systematically fill a database, especially for unexperienced mappers. Currently, there is a high possibility to apply mobile-GIS including GPS and mapping tools to significantly increase data collection efficiencies". Please explain the efficiency of the output. User obtains field data, not clear which geometries and contents. And which specific information is collected.

**Commentato [S6]:** A rapid offline-online technology is the output, absolutely appealing, but not well-defined as collect data on landslide events, hazard impacts and damaged infrastructure. Please specify which information and which aim users can collect during survey. This prototype provides a solution for preparing landslide hazard maps in relation with vulnerability. Too general, a Mobile-GIS offers support to landslide hazard with vulnerability (you did not introduce vulnerability before…). Be clear with details of aims.

**Commentato [S7]:** The chapter introduce risk management, with tasks and criteria. Landslide inventory is a part of methods available, but also the target of paper. What is the aim to illustrate all steps that you do not face? Focus on landslide inventory and the architecture provided for it.

[revised manuscript text omitted]

**Commentato [S21]:** Not clear

**Commentato [S22]:** ROOMA should improve quality and quantity of inventory. GPS is basically important, depend also by resolution and signal. Field survey usually requires control on GPS signal and calibration, otherwise field survey is not precis. Did you treat it? Which kind of satellite images do you use? Field data is corrected by images. But is it on field control or post-processing? Please specify this integration, it is fundamental.

**Commentato [S23]:** Title not acceptable. Integrate with previous chapter.

**Commentato [S24]:** Confusing. The previous methodology treats landslides with characteristics (materials, type, damage). Here element at risk. Merge all data type in same chapter.

**Commentato [S25]:** Do you mean landcover mapping? Otherwise you cite a company...

**Commentato [S26]:** Too vague, specify simply your aim.

**Commentato [S27]:** Which is the difference between Open Source Geospatial Software and Open-source geospatial technology. Clarify.

**Commentato [S28]:** Which DBS? Who is the owner.

**Commentato [S29]:** Why do you repeat so many times?

**Commentato [S30]:** This is innovative. You have to dedicate more time then past experiences on classic landslide database... Your app treats with PhoneGap, linked to existing web development. By website I read "hybrid applications built with HTML, CSS and JavaScript". You should specify which link on web storages, simply to include in your methodology.

files and transferred through the internet to the online component where the main database is located. This enables the collection of data from multiple data collectors to be entered into the same database. The geodatabase was designed to incorporate geospatial data acquired in the field, delivered as an input to the system (e.g., type, shape, volume, date, triggering factor, hazard degree) with elements at risk data connected to a specific event (e.g., building information, road network, damage information). The FOSS4G technologies selected to provide this module were PostgreSQL 9.4 (PostgreSQL, 2015) and Postgis 2.1 (PostGIS, 2015) for spatial database management. The GeoServer 2.6 (Geoserver, 2015) module, in connection with Geodatabase (Postgis), is delivered for spatial analysis and visualization. This component brings a complete and up-to-date description of the different layers including a landslide event layer, elements at risk layer and detailed information of landslides in the study area including event descriptions and photo clusters. Finally, the outcomes are captured and shown through GeoServer and OGC services such as Web Map Service (WMS) and Web Feature Service (WFS) as well as being exported as shapefile format and visualized in other GIS software like ArcGIS or QGIS. MySQL database (MySQL, 2015) and UserCake library (UserCake, 2015) improve the user management and authentication. Two type of users are available in the system: Public and Administrator. Based on their privilege, they can access to different components of the online version. For example, only the administrator can define a new study area and assign it to different users. Figure 5 displays the technologies and the frameworks of this prototype.

> **Commentato [S31]:** Which one?

> **Commentato [S32]:** Correct but redundant sentence

> **Commentato [S33]:** Describe which info by photo

> **Commentato [S34]:** User profiles

> **Commentato [S35]:** Not all components of architecture are explained. Consider them and introduce.

> **Commentato [S36]:** Why do you use prototype definition? The app will be updated, is not completed working or you need a piloting?

> **Commentato [S37]:** Explain which kind of combination.

The offline component of ROOMA (Figure 6) contains the following modules: 1. Geolocation, 2. Map with combination of multi-source base layer 3. Map drawer (Line, Polygon, Rectangle and Marker) 4. Satellite image as the base layer and 5. Saving options as Geojson-txt file in the offline mode. The mapping process is quick and easy; different features such as polygons, points or lines can be drawn on a map drawer after geolocation. Following, different satellite images as base layers assist for finding different objects on the map. However, the online component presents more modules besides map and geolocation modules: 1. Map with combination of multi-source base layer, 2. Saving online events directly to database, 3. Photo mapping, 4.Photo and event clustering, 5. User privileges 6. Data storage and analysis, 7. Import from/Export to Shape files.

> **Commentato [S38]:** Not clear which analysis you intend

The user can save or upload these features as one event and define additional characteristics such as land use, damage, trigger, possibility of hazard etc. Figure 7 and 8 illustrates how an admin can view different landslide events in the online version with the possibility of editing events.

> **Commentato [S39]:** Editing events of landslides based on satellite image is not innovative. Field survey exists since a lot. You should mark the online-offline technology as real advantage on field. You did not describe the relations to update database in online-offline condition. You should describe how can be data collected be synchronized. Do users choose online-offline mode or is automatic upgrade based on bandwith? Figure 8 is not innovative, simply you edit a polygon on a raster image, what is new?

**5. Study area**

Many landslide studies have been conducted in the Everest regions (Gupta & Saha, 2009; Bajracharya & Bajracharya, 2010; ICIMOD, 2016; Sato & Une, 2016). The 7.6 magnitude earthquake in Nepal on 25th April 2015 and a series of aftershocks significantly increased the risks of landslides (Collins & Jibson , 2015). Nepal has a high natural geological fragility which was further increased by the 2015 earthquake, which triggered several thousand landslides (ICIMOD, 2016; Collins & Jibson, 2015). The ROOMA application was tested in

the Phewa Lake Watershed (123km2) in Western Nepal, Kaski District (Figure 9) where our team has been monitoring landslides since 2013. An intense rainfall event (315 mm in 4 hours) killed 9 people on 29 July 2015 in Bhadaure-5 near Pokhara and another 25 people were killed nearby Lumle in Parbat District (BBC, 2015). It was very hard to differentiate those landslides and their properties through image interpretation so the urge for field mapping was very high and the landslides have to be identified on the field whether close to the event or far. The ROOMA application  run for a rapid assessment of landslides triggered by this event or reactivated along with their land-use characteristics and damages such as houses, schools, roads, rivers, agriculture fields and forest area. (Figure 10).

**Commentato [S40]:** Clarify distance, features, polygon revealed. Actually it is only a picture.

**6. Results**

, two days of field work were conducted in the Phewa Lake watershed, and based on medium resolution satellite image (GeoEye 2015, 5 meter resolution) added to ROOMA application, 59 landslides were mapped. The mapping of landslides (using polygons) was accompanied by data collection on land use features for each event (e.g. adjacent roads, rivers, forest, and critical infrastructure) to give better indications of surrounding features. The extreme advantage of mobile-GIS is gained in relation to the existence of landslides and determination of the frequency distribution of landslide areas. The satellite image added to the application significantly eased the exploration of this area and assisted the visual interpretation process. The data were collected on-site either close to road or from a distance which enabled easy interpretation for landslides which would have been difficult to access otherwise (Figure 11 and 12). Figure 11 represents a new landslide documented near the road that was not visible in satellite image and  Figure 12 shows a larger landslide which was located within a distance and it is clearly visible in image interpretation. Most of large landslides were mapped by distance. Figure 13 shows the distribution of landslides in that area where most landslides occurred in the centre.

**Commentato [S41]:** Parameters not present in previous list within methodology. Did you add new text? Why?

**Commentato [S42]:** Clarify the link between mobile-GIS and frequency distribution

**Commentato [S43]:** It is a reason why you integrate satellite image. It has to be mentioned within definition of methodology

**Commentato [S44]:** It is a bit ambiguous. You update landslides with you field actions, but some of events are not accessible, but visible only with distance like in Figure 12. I would consider as integration.

**Commentato [S45]:** Large landslides are visible on satellite. Did you edit on desktop GIS or check shape of landslide on field. IT could be a tool to upgrade what is existing as polygon.

All data were uploaded to the online version and then exported to a shape file. It was possible to perform the rest of the analysis in QGIS however it is planned to add extra modules in online version for querying, summarizing results and finally having landslide susceptibility map. Data obtained from the field survey were successfully analysed in the Open Source GIS with more detailed analysis possible such as distribution of landslide type, material, elevation, damages, surface areas and volume, graphics production, spatial modelling, and visualization of many types of data. For example, all the information about land use characteristics and their damages for different landslide were gathered separately in our database and can be useful for more detailed analysis.

**Commentato [S46]:** These are scientific and practical results. Not clear and too general. Delete.

**Commentato [S47]:** New version of output

Moreover, further analysis of land use/cover changes has been carried out based on visual interpolation on a multispectral satellite image (SPOT 2016, 2 meter resolution) acquired in 2016 after this field checking. Basically our ground truthing brought the confidence for further mapping (177 Landslides mapped afterward)

of the additional smaller landslides that were not mapped during the field survey. Figure 15 shows these landslides on the map.

The advantage of a mobile version in field over mapping using only GIS and high resolution satellite images (in office), is that some features characteristics of landslides are not visible only on images. Coupling satellite image interpretation with field observation allow to identify better the type of landslide, even using a medium resolution satellite image (~5 m). The Figure 16 shows such example: the detail mapping on standard GIS permits to identify active landslides in the gullies, i.e. debris-flow and shallow landslides, while the lower resolution image coupled with field survey permits to identify a larger landslide. The landslides linked with the gullies is simply the limits of the larger one, where the activity is obvious.

**Commentato [S48]:** This conclusion is positive. You declare landslides in gullies visible with high resolution images, while adding filed survey you can define and edit better the polygons.

**7. Discussion**

Landslide inventories define vulnerability, hazard, landslide susceptibility and risk by investigating the information on type, pattern, distribution and slope failures (Guzzetti, et al., 2012). Earlier works on landslide hazard evaluation shows that considerable developments have been accomplished in the last decade, GIS tools are now crucial for landslide hazard and risk assessments, however, the generation of landslide maps including elements at risk and an online database in a larger scale appears a stage too far especially in data poor countries having such an offline application can provide a significant technological leap and save valuable resources. The value of landslide inventories relies on the accuracy and certainty of the information which is problematic to define (discussed in introduction) however, different mapping approaches on Open Source Geospatial technologies, can significantly simplify the production of these maps.

**Commentato [S49]:** Reply all these aims within discussion. While you focus simply on data collection and database...

**Commentato [S50]:** Quite old sentence...

**Commentato [S51]:** You repeat several times elements at risk, but I do not see some examples about field trip or database about them.

This application incorporates rapid, economic and participatory methods for mapping landslides. It uses satellite images as multi-source map and enables multiple data collection to finally be collated in a centralized database. Data can be acquired in offline version using android device or an online mode using all browsers in Pcs, tablets and mobiles. The study was applied for mapping landslides in post-earthquake Nepal, but, it can be practical for other hazard events such as floods, avalanches, etc. Nevertheless, this offline version can be improved by adding more components for distance calculation, continuous lines sketch, recording foot paths and merging the GPS located camera with the azimuth of data to help generating 3D models of the area.

**Commentato [S52]:** You do not mention before. Who participates and which organizations or teams.

Considering all the difficulties stated in this work, a landslide mapping are typically carried out based on the experience of the expert however, by getting support of mobile GIS, this application is easy to be run by non-expert and general public as well. A combination of satellite data and web-GIS technologies brings the ideal solution for landslide hazard and risk data acquisition especially more high resolution satellite images can be available recently and sometimes freely. The paper concludes that the ROOMA tool will increase the quality

**Commentato [S53]:** Concept of data synchronized and gain of time with ROOMA are not mentioned.

of landslide maps as well as , and landscape modelling and will also assist the speed for preparation of above products.

The paper accomplishes several of new improvements and future works, for example adding the topographic data DEM, spatial-temporal modelling by using landslide inventory maps. More works are needed to incorporate vulnerability components, where more attentions are needed in defining vulnerability values in order to generate risk maps. Finally, it is essential to integrate a spatial decision support systems to use such data for landslide hazard and risk assessments for both stakeholders and local authorities.

---

## Author Comment (AC1) · 23 Oct 2016

Dear Reviewer,

We would like to thank you for the comments and also your time to review this paper carefully with details. In supplement, you can find the answers regarding your questions and comments in "Answer.pdf". The corrections have been prepared in "Manuscript.v4.pdf" that is also attached in Supplement. Those in red are the updates. This is not the final version of the Manuscript. We will update the final version, after receiving the other review too. Thank you again for your time and effort and wishing you a great day.

[Figure]

Sincerely, Roya Olyazadeh on behalf of the other co-authors

Please also note the supplement to this comment:
http://www.nat-hazards-earth-syst-sci-discuss.net/nhess-2016-267/nhess-2016-267-AC1-supplement.zip

———————————————

---

## Referee Comment (RC2) · Anonymous Referee #2 · 20 Dec 2016

"An Offline-Online WebGIS Android Application for Fast Data Acquisition of Landslide Hazard and Risk" deals with an interesting and innovative topic, that is mobile tools for field landslide mapping. In particular the authors developed a prototypal App which enables the visualization of several cartographic satellite maps used as background layers upon which the user can draw the contours of the landslides recognized in the field. The App is able to upload data to a database once an internet connection is available and to export the products as shapefiles. The main issue with this paper is clarity. First of all the architecture of the system is not clear and a figure showing it is also missing. I suggest to better to explain the temporal process involved in the publication of the offline data. In particular what parts of process are automated and

what are manual. Software with deferred updating may have problems when are used by more than one user. For example there may be problems due the digitization of the same landslide by two different users. What solutions have been adopted to solve this issue? English is often incorrect or not fluent to the point that only some of the major points have been signalled (see my specific remarks below). Punctuation should also be revised. Therefore I recommend that the paper undergoes a professional English check. Some figures also are often not completely clear or even contain errors. See the specific remarks below. When citing more than one reference in the text be sure that they are sorted following the criteria of NHESS. All considered I recommend major revisions.

Specific remarks: Page 1 line 2: remove the semicolon. P1 l14: what do you mean by "complications subject to accessibility and terrain"? One of the advantages of remote sensing is indeed to overcome accessibility issues. P1 l16: add "the" before "implementation". P1 l19: replace "for instance" with "such as". P1 l20: remove "of". Also (here and elsewhere), PostgreSQL and PostGIS are cited like two separated product. It is better to report PostgreSQL as the real DBMS and PostGIS as its plugin for spatial database management. P2 l2, l7, l8, l9 etc: when citing references do not put an empty space before comma and do not use comma before "et al." (here and elsewhere in the text). P2 l7: "selection of techniques relies on". Not clear, please rephrase. P2 l11: replace "are" with "is". P2 l15: replace "have long" with "require a long". P2 l16: check English. P2 l18: replace "disadvantages" with "issues/problems". P3 l7: improvement with respect to what? Please clarify in the text. P3 l23: when references are cited within a sentence only the dates must be in the brackets. P4 l15: Another methodology that should be referenced in this paragraph is data mining from newspapers. P5 l12: BGS Sigma is reported as 2013 in the reference list and further in the text. P6 l4-5: these two sentences are not connected with the following of the paragraph. P8 l29: the data transfer system between offline program and online component is not clear. Is it a normal web application? the data sending is automated or the user must select it manually from his device? P8 l35: It's not clear the role and the position of

GeoServer and PostGIS DB (are they in a remote server? in the same server or in two separated servers? maybe a figure about System Architecture could help). Then it's not clear if GeoServer is used only as map server or if it is used also to receive the GeoJSON made from the mobile app (through native REST API or WFS protocol) or if this is done by another component connected to PostgreSQL/PostGIS. P9 l5: Is there a technical reason to use two different DBMS (MySQL and PostgreSQL) in the same project? (again a System Architecture figure could help). P11 l7: replace "out team" with "the authors". P12 l7-8: this sentence it unclear. Please rephrase. P12 l14: in the centre of what? P17 l3: here you have started the description of ROOMA concerning its database and then you talk about the test site. Before you start talking about the test site finish the description of ROOMA, talking about the offline drawing tool, the possibility to upload data and to export data in GIS format. P17 l12-13: please be more conservative in this sentence, i.e. instead of stating that ROOMA will increase the quality of landslide maps, state that this it its aim, or that it provides a contribution in that direction. P17 l14-15: if this paper accomplishes something or not should be left to the reader to decide. Furthermore here you say that the paper accomplishes something that is still to be developed. Please change this sentence.

Figure 2: what do you mean with "temporal"? The state of activity? Also, in the central box remove the capital letter from "L" in "Slope". Figure 3: this figure should be changed into a table. However in its present form it is very confused and confusing. Most notably, the second column should show names of techniques but also reports "frequency", "earthquakes" and others. Also, what are "exicting data"? Figure 4: why in the landslide information are also here (left box) even though this concerns information concerning the elements at risk? Are not landslide information already contained in the landslide database (figure 3)? Please explain. Figure 10: in the caption replace "so many" with "several". Figure 14: please add what is on the Y axis. Also the subdivision of a column between "feature" and "damage" is unclear. What do you mean by feature? This must be better explained. Table 1: It is not clear if the fields that you report here represent all the possible entries of your App or if there are just some reported as examples. In

the first case I suggest to add the actual interface of your App showing how filling in the landslide database works. Instead of "numbers of landslides" state "progressive identification number of the landslide". Also, why is it written "initiation" within the types of movements?

---

## Author Comment (AC2) · 13 Jan 2017

Dear Reviewer, We would like to thank you for reviewing this paper, your time and effort for corrections and suggestions. Attached "Supplement: nhess-2016-267-supplement.pdf " you can find our answers to your comments. Base on your request, we have updated the section on technology and architecture. We also have updated some figures such as data model and flowchart for more clarification.

Best Regards, On behalf of all Authors, Roya Olyazadeh

Specific remarks: Page 1 line 2: remove the semicolon. Updated P1 l14: what do you mean by "complications subject to accessibility and terrain"? One of the advantages

of remote sensing is indeed to overcome accessibility issues. RS are mostly used because of the accessibility and terrain. So of course that is an advantage of RS. P1 l16: add "the" before "implementation". Updated P1 l19: replace "for instance" with "such as". Updated P1 l20: remove "of". Also (here and elsewhere), PostgreSQL and PostGIS are cited like two separated product. It is better to report PostgreSQL as the real DBMS and PostGIS as its plugin for spatial database management. Updated P2 l2, l7, l8, l9 etc: when citing references do not put an empty space before comma and do not use comma before "et al." (here and elsewhere in the text). The references were added using REFRENCES in word 2010. We have updated them all according to NHESS. P2 l7: "selection of techniques relies on". Not clear, please rephrase. Updated P2 l11: replace "are" with "is". Updated P2 l15: replace "have long" with "require a long". Updated P2 l16: check English. Updated "Landslides are often small, with a high frequency of occurrence and located in remote areas which are difficult to access." P2 l18: replace "disadvantages" with "issues/problems". Updated P3 l7: improvement with respect to what? Please clarify in the text. Updated as follows: "Data management improvement in hazard event mapping and storage using new technologies such as Postgis and Geoserver." P3 l23: when references are cited within a sentence only the dates must be in the brackets. Updated P4 l15: Another methodology that should be referenced in this paragraph is data mining from newspapers. This figure related to techniques for landslide data acquisition and section has been removed from the paper as requested by the previous reviewer. P5 l12: BGS Sigma is reported as 2013 in the reference list and further in the text. Updated P6 l4-5: these two sentences are not connected with the following of the paragraph. We have deleted those and we also updated the methodology with figure 3. P8 l29: the data transfer system between offline program and online component is not clear. Is it a normal web application? the data sending is automated or the user must select it manually from his device? We have updated it. The transfer is done by uploading Geojson files from android app to the online version. The online version is a normal web-mobile browser. The data are not being sent to the server automatically and the user has to upload those Geojson files

from the device to the online system. The upload is available in the online version and can be done by one click. P8 l35: It's not clear the role and the position of GeoServer and PostGIS DB (are they in a remote server? in the same server or in two separated servers? maybe a figure about System Architecture could help). Then it's not clear if GeoServer is used only as map server or if it is used also to receive the GeoJSON made from the mobile app (through native REST API or WFS protocol) or if this is done by another component connected to PostgreSQL/PostGIS. Yes, our server was based in Geneva and we were working in Nepal. We used one server however, Postgis and GeoServer, which were installed separately in our system. Geoserver is used as map server and to download the final data into shpfiles. We have updated the technology section with more detailed information.

P9 l5: Is there a technical reason to use two different DBMS (MySQL and PostgreSQL) in the same project? (again a System Architecture figure could help). Yes, we used MySQL for user management which was a ready open source package called User-Cake, So we did not have to program and deal with user management separately and for spatial database we used Postgis which is under Postgres database. As mentioned in the text: P9 l5 6 : MySQL database (MySQL, 2015) and UserCake library (UserCake, 2015) improve the user management and authentication. P8 l33 34 : PostgreSQL 9.4 (PostgreSQL, 2015) and Postgis 2.1 (PostGIS, 2015) for spatial database management. P11 l7: replace "out team" with "the authors". Updated. P12 l7-8: this sentence it unclear. Please rephrase. Updated: The advantage of mobile-GIS is increased in relation to the existence of landslides and distribution of landslide areas. P12 l14: in the centre of what? In the center of our case study, Pokhara Lake Watershed. We have updated the text accordingly. P17 l3: here you have started the description of ROOMA concerning its database and then you talk about the test site. Before you start talking about the test site finish the description of ROOMA, talking about the offline drawing tool, the possibility to upload data and to export data in GIS format. We have updated accordingly. P17 l12-13: please be more conservative in this sentence, i.e. instead of stating that ROOMA will increase the quality of landslide maps, state that this it its

aim, or that it provides a contribution in that direction. Updated: "ROOMA tool aims to increase the quality . . ... " P17 l14-15: if this paper accomplishes something or not should be left to the reader to decide. Furthermore here you say that the paper accomplishes something that is still to be developed. Please change this sentence. Updated Figure 2: what do you mean with "temporal"? The state of activity? Also, in the central box remove the capital letter from "L" in "Slope". This refers to landslide maps for different time periods. It is a cited figure. We have updated "Slope". Figure 3: this figure should be changed into a table. However in its present form it is very confused and confusing. Most notably, the second column should show names of techniques but also reports "frequency", "earthquakes" and others. Also, what are "exicting data"? This figure is removed from the paper and it has replaced with the following figure to give a better idea of how this application can work.

Figure 4: why in the landslide information are also here (left box) even though this concerns information concerning the elements at risk? Are not landslide information already contained in the landslide database (figure 3)? Please explain. We have removed figure 3 and 4 as also requested by reviewer 1 and we updated it with a data model. Previous figure 4 was to show how the database could be used in our server by separating spatial and non-spatial data. We also updated the element at risk paragraph and added it to a previous caption.

Figure 10: in the caption replace "so many" with "several". Updated as requested Figure 14: please add what is on the Y axis. Also the subdivision of a column between "feature" and "damage" is unclear. What do you mean by feature? This must be better explained. The Y axis is number of Landslides, we have added it accordingly to the figure. Features are for each landslide, for example 56 landslides occurred in forests and of these, 44 damaged the forest (red = Damage). We also added more description for clarity in the figure. We have updated the figure as follows:

Table 1: It is not clear if the fields that you report here represent all the possible entries of your App or if there are just some reported as examples. There are the possible

entries for our app. We also added the data model in the new version for more clarity.

In the first case I suggest to add the actual interface of your App showing how filling in the landslide database works. Instead of "numbers of landslides" state "progressive identification number of the landslide". Also, why is it written "initiation" within the types of movements? Filling the landslide database is conducted in mobile form (Table1). We will update the interface figure 6 by adding this form to the figure. We have removed initiation from types of movements.

Please also note the supplement to this comment:
http://www.nat-hazards-earth-syst-sci-discuss.net/nhess-2016-267/nhess-2016-267-AC2-supplement.pdf
* * *
[Figure]

Figure 3: this figure should be changed into a table. However in its present form it is very confused and confusing. Most notably, the second column should show names of techniques but also reports "frequency", "earthquakes" and others. Also, what are "exciting data"?

This figure is removed from the paper and it has replaced with the following figure to give a better idea of how this application can work.

[Figure]

Figure 3: Workflow of ROOMA

**Fig. 1.**

Figure 4: why in the landslide information are also here (left box) even though this concerns information concerning the elements at risk? Are not landslide information already contained in the landslide database (figure 3)? Please explain.

We have removed figure 3 and 4 as also requested by reviewer 1 and we updated it with a data model. Previous figure 4 was to show how the database could be used in our server by separating spatial and non-spatial data. We also updated the element at risk paragraph and added it to a previous caption.

[Figure]

**Figure 4 Data model of ROOMA: Database is automatically created from Geojson-txt files which have been uploaded into online version of ROOMA.**

**Fig. 2.**

In the first case I suggest to add the actual interface of your App showing how filling in the landslide database works. Instead of "numbers of landslides" state "progressive identification number of the landslide". Also, why is it written "initiation" within the types of movements?

Filling the landslide database is conducted in mobile form (Table1). We will update the interface figure 6 by adding this form to the figure. We have removed initiation from types of movements.

[Figure]

Figure 6 : Offline Component with a satellite image as a background: Geolocation (Geo), Stop Geolocation (ST), Show all the attributes in a pop up window (Pp), Reset the map (RE) Save as GeoJSON-TXT (SV) by filling the green from.

**Fig. 3.**

Figure 14: please add what is on the Y axis. Also the subdivision of a column between "feature" and "damage" is unclear. What do you mean by feature? This must be better explained.

The Y axis is number of Landslides, we have added it accordingly to the figure. Features are for each landslide, for example 56 landslides occurred in forests and of these, 44 damaged the forest (red = Damage). We also added more description for clarity in the figure. We have updated the figure as follows:

[Figure]

**Figure 14: Relationship between features and landslides damage:** for example 56 landslides were happened in forest and 43 out of 56 damaged the forest (Red = Damage).

**Fig. 4.**

---

## Author Response (AR1)

Dear Editor and Referees,

We would like to thank you for reviewing this paper and also your time for corrections and suggestions. We have revised the new version based on the comments of reviewers. Following are the list of main changes in the revised version. The next two parts are the corrections for reviewer 1 and 2. The last part shows the marked-up version with all changes in red.

Again many thanks and we are looking forward to hearing from you.

Best regards on behalf of all authors,

Roya Olyazadeh

List of main changes:

1. Landslide maps are replaced with Landslide Inventory Maps (LIMs) (Requested by review 1).
2. Background is updated (Requested by reviewer 1).
3. Methodology has been renamed to Implementation and merged to just one section and data model is added (Requested by reviewer 1 and 2).
4. Architecture and more details about technology used are added to Technology section (Requested by reviewer 1 and 2).
5. Conclusion section is updated.
6. Some more References are added and then updated based on NHESS (Requested by reviewer 1 and 2).
7. Figure 3 and 4 are removed and replaced with new figures for better understanding (Requested by reviewer 1).
8. Figure 6 and 14 are changed and updated (Requested by reviewer 2).
9. More changes are added in the paper and it is updated regarding English correction and grammar.
10. Other changes can been seen in revised version which have been directly answered under their comments.

**Review 1:**

I was glad to review your paper, interesting for contents and final aims. I list hereafter tasks you should review for publication. I will reconsider your paper after major revisions. The argument is encouraging, but actually incomplete for methodology and output. Specific reviews of chapters appears hereafter.

Paper is written with current and regular languages, you adopt technical jargons but not with accurate quality of single chapters. Introduction and background list a context in which the paper reveals the output. Concerning landslides, you introduce susceptibility, hazard, risk, survey types, patterns, distributions, statistics of slope failure, management strategies, risk assessment… Monitoring systems miss in the list, to complete landslide treatment from geomorphological and geophysical point of view (too much…) The general description is quite systematic but the references are not complete, because topics are vast, complex and only cited, often with a general redundancy.  The database chapter explains technologies for data collection and GIS system for inventory of landslides in a territorial context. The methodology chapter should be the core, but it lists data and characteristics required for single landslide, integrating characteristic of landslide with element at risk, unclear and indefinite. Technology and Platform appears as core of the output with a copious integration of know-hows and available solutions. The study area illustrates landslides with subsequent results, but without specific enlightenment. Susceptibility map with spatial modelling and many types of data emerge again within chapter, confusing the real output.

The paper contains an innovative issue but not well ordered. ROOMA has a complex architecture, for gathering and field survey. The activity includes android environment to deploy free and open solution for data collection. The idea is offering a crowd system, to combine user-friendly tools for geospatial activity on field. The participation could include contribution in a large area, with rapid methods for slide mapping. This challenge is innovative, but not included as priority in the paper, because not clearly linked to the landslide dataset required. The paper overbalances interest on all details of risk assessment, deleting request of landslide data and technology adopted.

Digital field survey exists since around 10 years, within geomorphology final aims, controlled by high precision in GPS location and field GIS integration (MapIt, ArcPad, Geopaparazzi, GISTrimble, and other FOSS4G solutions). You mentioned tablet and mobiles, but the advantage of platform is android environment, customizable and free of costs. The advantage is the online-offline, independent by bandwidth, offering a tool definitely fast, user-friendly and low-cost. These advantages are not enlightened. Clear problems could be bug-fixing, GPS precision. The advantage of online-offline includes clear benefits, you need to highlight them compared to traditional field survey.

You explain the aims as android mobile application on both Offline-Online access. The aim is a fast and storing of data. The visualization and drawing tool is based on central database available to services (mobile, PCs and web browser). Data management improves in hazard event mapping as you declared, but the aim is too general, not simply split.

Mobile-GIS has a clear gain, but limits due to dimension of mobiles, resolutions, spatial tools available, zooming, spatial extent, route, snapping and editing tools have to be revealed. If compared to desktop GIS you have to explain the difference. ROOMA has a mobile solution, introduced in data transfer. Specify the content of slides and clarify what users can do on the field. Mobile-GIS with GPS are tools to increase efficiency in data collection.

Online-offline is an interesting approach with Geojson. How can data be saved and furthermore included in geospatial analysis? Explain a bit better the technology adopted. The architecture is not well shown.

Dear reviewer, regarding you general comments we would like to add the following points:

1. In this work, we do not consider any monitoring system. The idea is to map landslides fast and easy using mobile field survey and satellite image in the same time. Besides to create a database that is also easy to update as the offline can be easily connected to online by geojson-txt file. So even the uploads are very simple.

2. This application was tested in the field with ITC (Netherlands), Yale-NUS College (Singapore), University of Kathmandu and was requested by ICIMOD (Nepal), UNEP, Canton Vaud for forestry in Switzerland, and University in Tunisia. They were all interested to use this application and requested for an updated version. Due to our limited time, we have provided an updated version only to Singapore, Nepal, and Canton Vaud for testing. Recently, YaleNUS College and Canton Vaud have tested the offline application and they provided their feedbacks and comments on the application which some mentioned in the last chapter. We believe this application is far beyond the digital field survey because abovementioned showed their interest to use this app.

3. The advantage of this application was mentioned through the whole paper. We talked about free of cost, fast and storing in central database, following mentioned some. We have updated the new version with more highlight in the advantages of this application.
   a. "Abstract: can take advantage of Open Source web and mobile GIS tools for an improved ground-truthing of critical areas."
   b. "Abstract: This prototype assists for quick creation of landslide inventory maps (LIMs) by…"
   c. "Introduction : 2. Fast and easy acquiring  and storing of data and information"
   d. "Methodology: This methodology compensates the lack of landslide inventory and precise topographic process, decreases the resources and time needed for storage and update. In addition, the combination of the ROOMA data collection method in the field with GPS and satellite image as source maps can significantly improve the accuracy of input field data."
   e. "Technology:    The mapping process is quick and easy."
   f. "Conclusion: Moreover, the ability to use the Open Source software indicates that analyses can be carried out without incurring the high costs associated with software acquisition, a particular advantage for developing country, researchers and government officials."

4. We mentioned about the traditional methods (As your request on S10 to remove the different technique, the figure is removed and replaced with figure 2.) but comparison with them is not the purpose of this work. Indeed we wanted to show the differences of the work has been done in the office compared to our field work, which is highlighted in the result.

5. References for this paper were 48. We have added some more references as  you requested that references were not complete

6. "Database" sub-chapter was connected to the background and talking about available database around the world. As it was not clear for reader, we merged it with subchapter Landslide data collection. We also added more details and a figure for data model of this study in methodology

7. The ROOMA has a complex architecture, that is true, but gathering the data and field survey for ROOMA is not complex and it is very easy. In our field trip, the data was recorded by those who did not have proper experience on using tablet or android application and they found it very simple. It just took 15 minutes in the field to show them how they can use this app and the rest of the 2 days of our field trips, they mapped all the landslides without any issues. We did not do any comparison test about field work and office work regarding gain of time however we did compare them by result. We tried our best to simple the offline version. The online version however needs more knowledge and experience especially for an admin user.

8. The tutorial and codes will be updated at the end of the study within a university link. For the technology adopted and database we have added more details and figure of data model in the new version.

9. Regarding your several questions about synchronization of the data: Data are saved in Geojson-txt file and then uploaded to the database when internet is available (as mentioned in the paper) so there is no synchronization. The developer should know how to extract data as geojson out of Leaflet map and then transfer it to a file using php and finally using another script in PHP, having the option to upload them back to server again. These are very technical and as mentioned tutorial and codes will be available later. We tried our best to explain it which NHESS readers can understand it simply.

We have merged and updated the structure of the paper as you requested. Following you can find your answers regarding your separate questions in the text.

Commentato [S1]: Please simplify differences, is not clear for data typology and requirements.

We have deleted this sentence and we have updated "Landslide map" to landslide inventory maps (LIMs).

P.2 line 4:  LIMs are important factors for …..

P.3 line 9: …with a field survey for preparation of LIMs in relation with elements at risks.

P.16 line21:  This result illustrates that using this platform will raise the quality of LIMs, including susceptibility…

And some more.

Commentato [S2]: The access of field area is difficult (dynamic nature, what is?), because you require landslide inventory for final map.

The dynamic nature of landslides refers to the danger and difficulty to access and measure, where landslides usually happens in steep area with the possibility of reactivation. This is a cited sentence.

Commentato [S3]: Long process and intensive resource. What are these parameters?

P.2 line 14: We have updated to: "Landslide inventories are time consuming and resource intensive".

Here again, it is a cited sentence to explain the difficulty of making landslide maps and mentioned in the text as follow:

"Typical issues for creating these maps include (Guzzetti, et al., 2012; van Westen, et al., 2006; Safaei, et al., 2010):"

Commentato [S4]: "GIS for landslide susceptibility and hazards" is redundant in the paper. These are complex and fundamental steps in risk assessment, with methodology since 10 years… Why do you consider these measurements as concept linked to landslide inventory?

In the new version, we tried to reduce them. As it is mentioned in the background and figure 1, landslide inventory serves as the basis of landslide susceptibility, hazard and risk. They all need a data collection or verification process in the field.

Commentato [S5]: "mobile-GIS offers technology for more effective ground-truthing and a rapid tool which can systematically fill a database, especially for unexperienced mappers. Currently, there is a high possibility to apply mobile-GIS including GPS and mapping tools to significantly increase data collection efficiencies". Please explain the efficiency of the output. User obtains field data, not clear which geometries and contents. And which specific information is collected.

The user can obtain any kind of data in the field with GIS and GPS mobile technology. The GPS is given 4 to 15 meter accuracy. However, as we also use satellite image as base layer, the accuracy of the map depends on the quality of that image. In this work we used a 5 meter resolution satellite image (explained in result chapter). Coupled field survey and image interpolation, definitely increase the quality of our data. The detailed information of which data collected can be seen in the next paragraph and chapter 3. We updated as follows:

P.2 line 27 : "Currently, there is a high possibility to apply mobile-GIS including GPS and mapping tools to significantly increase data collection efficiencies such as location accuracy and detailed information of features."

Commentato [S6]: A rapid offline-online technology is the output, absolutely appealing, but not well-defined as collect data on landslide events, hazard impacts and damaged infrastructure. Please specify which information and which aim users can collect during survey.  This prototype provides a solution for preparing landslide hazard maps in relation with vulnerability. Too general, a Mobile-GIS offers support to landslide hazard with vulnerability (you did not introduce vulnerability before…). Be clear with details of aims.

The application with online-offline technology was tested in a real study area for data collection of landslide events (figure 13), hazard impacts and damaged infrastructure (figure 14, 15, 16), which is the three main outputs of the application. The aim is to facilitate the data collection process in the field using this advanced technologies for authorities, stakeholders and the general public. The detailed information of which types of data/information are collected can be seen in chapter Result and Implementation.

Since the paper is focused on the application, we only added some selected results and comparison with the old version of visual interpolation (figure 15, 16). We have added this following sentence (more details of outputs can be seen in the result chapter):

P.2 line 31: "The preliminary result of this application is also compared to the results obtained from satellite image interpolation."

We have deleted the vulnerability sentence to avoid complication and added this:

P.2 line 30: "An offline technology helps to map the events, especially in rural areas where internet is not available."

Commentato [S7]: The chapter introduce risk management, with tasks and criteria. Landslide inventory is a part of methods available, but also the target of paper. What is the aim to illustrate all steps that you do not face? Focus on landslide inventory and the architecture provided for it.

This is the background chapter for landslide hazard and risk which is a part of the title. We mentioned available methods and why landslide inventories are more important, serving as the basis of landslide hazard and risk assessment and are the simplest method. In the sub-chapters of this background chapter, we focused more on the background technology and landslide inventories which you also suggested to remove them in S10.

We have deleted following sentence as we did not use them further:

"The classification comprises three different methodologies: 1. Qualitative 2. Semi-quantitative and 3. Quantitative."

Commentato [S8]: Which kind of stakeholders and for which role?

We have deleted stakeholders from the sentence for clarity. This paper and work does not focus on stakeholders and roles.

P3 line 13: "Landslide risk management estimates risk options with different levels of acceptance criteria."

Commentato [S9]: The subchapter is poor of matters. You list again landslide inventory data, hazard factors, and elements at risk with a table about contents, but you declare to focus on inventory one. Why do you need to repeat? I suggest merging with next chapter.

We have merged as you suggested This sub chapter has been merged to Landslide data collection.

P.4 line 17: "Historical landslide records and freely accessible databases have been developed for a few countries, (e.g. Italy (Guzzetti, 2000), Switzerland, France, Hong Kong (Ho, 2004), Canada and Colombia). However, difficulties related to completeness in space and time are a drawback (van Westen et al., 2006)."

Commentato [S10]: Techniques for data collection are actives since years. The list of them is outside the goal of your work.

The title of this paper is "Fast Data Acquisition of Landslide Hazard and Risk" and we want to highlight how the mobile-GIS technology plays an important role in acquiring ground-based field data collection. Therefore, we believe that it is important to mention different/conventional and available techniques to prepare landslide inventory maps (also as you requested in S7). Subsequently, we mentioned in the paper that there are few works using mobile technology for landslide field survey, which served as a motivation of our work.

We have deleted the figure 3: "Overview of techniques for landslide data acquisition" as you requested.

Figure 3 is updated as follow:

P.6 line 23: "Figure 3: Workflow of ROOMA where coupled image interpolation with field survey leads to asset of maps and complete database of landslide data and their characteristics. These different maps of landslide distribution, hazard, and damage infrastructure can be produced by manipulation in GIS."

Commentato [S11]: You reveal existing background on similar experiences, but a bit poor. Please enlarge examples, here a bit limited.

We have updated this sub-chapter: Mobile and web GIS for landslide inventory   Please refer to page 5.

Commentato [S12]: What is the aim of this sentence? It is out of context.

We have deleted. Please refer to page 5.

Commentato [S13]: Please explain what is

BGS-SIGMA is the name of the application. For more details you can refer to the resource. We have updated more details.

P.5 line 3: "The BGS digital field mapping system (BGS-SIGMA mobile 2012) includes customises ArcMap 10 and Ms Access 2007. **It is customised of two toolbars for mobile and desktop.  The mobile toolbar** is to capture the data in the field on rugged tablet PCs with integrated GPS units **and desktop toolbar focuses on data interrogation, data interpretation and the generation of finalised data.  This is free software however it requires** Arc Editor Licence to run (BGS, 2013)."

Commentato [S14]: If your mark Open Source GIS as guideline, these techniques does not look compatible…

We did not use any of these techniques. It is just an example of different GIS tools for landslide inventories or data collection. We will update this chapter with more details of why each was not suitable for our study.

Commentato [S15]: Output running on rugged tablet, able in few copies, only for technicians. Not clear why it is not in Technology of ROOMA.

GIS technologies are wide, and most of the times, they are selected based on the developer's preferences, capacity, knowledge and ease of use. In our case, we did not select ArcGIS and ArcMap as (obviously) they are not free and open source solutions. We updated this in the respective chapter.

P.5 line 22:  "All the above mentioned systems have some disadvantages for our study such as: limited access (BGS, 2013), limited drawing tools (GeoData, 2015) (e.g. point markers only), desktop GIS (Mantovani et al., 2010; Acharya et al., 2015), paperfield systems (Temblor, 2016), and limitations related to visualization and data acquisition (UNEP, 2014)."

Commentato [S16]: Mobile app in android collects info on field, I supposed by different users. This is the advantage. Dot point or polygons can be marked on field since a lot of years. MapIT e.g. or ArcPad, if you maintain ESRI environment. If you pass on Open Source GIS the environment is another. With other solutions.

**MapIt** is no longer available for purchase. The database capabilities of the Spatial Data Service (SDS) in MapIt will be available through ArcGIS for Server Basic version 10.1. [ref: http://www.esri.com/software/mapit)]

MapIT is no longer available and ArcPad does not look compatible for this paper (as you mentioned in S14) because they are not open source and we do not focus on digitizing in this chapter. This chapter rather explains available frameworks and platforms for data collection related to landslide data collection and available online database for landslide hazard and risk. The purpose of this work is far beyond the digitizing, however, drawing tools plays an important role in this application because we made it extremely easy and fast. We added the advantage of this mobile application in the result chapter.

Commentato [S17]: You mentioned GeoServer, here you list only MapServer. Why this choice?

This was simply a reference of the cited works, in which MapServer was applied. In this Chapter, we did not compare different technologies and it was rather focused different available platforms on using GIS for the landslide inventory. We mentioned already, but we will update with the clarification of why we could not use any available platforms and we implemented our own platform.

Commentato [S18]: Cadaster?

A cadastre (also spelled as cadaster), https://en.oxforddictionaries.com/definition/us/cadastre.

P.5 line 18: "for data collection of cadastre (cadaster) mapping".

Commentato [S19]: Two professional outputs, one is a company and one is a crowd emergency webgis. What is your choice?

We explained different available platforms for GIS landslide and the only one using mobile was BGS-SIGMA. So we provided some other popular mobile applications which are not about landslide but they are all using mobile for data collection in the field. However they were not still advantageous for our works as we needed satellite image and offline version working together. Therefore none were compatible to our work that is why we made a new application. We have updated it at the end of this chapter.

P.5 line 22 to 27: "All the above mentioned systems have some disadvantages for our study such as: limited access (BGS, 2013), limited drawing tools (GeoData, 2015) (e.g. point markers only), desktop GIS (Mantovani et al., 2010; Acharya et al., 2015), paperfield systems (Temblor, 2016), and limitations related to visualization and data acquisition (UNEP, 2014). There are different systems in mobile GIS and data collection; however, the possibility for having an open-source- mobile application, with an added satellite image in offline mode, precise mobile GPS, easy and fast drawing tools, advanced visualization, and database management system, for landslide data collection is quite necessary."

Commentato [S20]: Why do you collect point, line and polygon as shape of landslides? Do you plan different methodologies?

No. Landslides can be collected on all different shapes and we mentioned before that the best practice is polygon. Sometimes, collecting data in the fields are in urge and therefore, they do not have time to draw polygons. In this case, they can simply use a point marker or maybe a line to record it, and then they can update and edit it later in the office. This is a user-choice depending on their needs and the application made it possible for their preferences.

Commentato [S21]: Not clear

Updated as:

P.5 line 31: "This approach compensates the lack of landslide inventories and precise topographic process, and decreases the resources and time needed for data storage and updating."

Commentato [S22]: ROOMA should improve quality and quantity of inventory. GPS is basically important, depend also by resolution and signal. Field survey usually requires control on GPS signal and calibration, otherwise field survey is not precis. Did you treat it? Which kind of satellite images do you use? Field data is corrected by images. But is it on field control or post-processing? Please specify this integration, it is fundamental.

We mapped it directly on the field and therefore, we did not do any post-processing afterwards. We using mobile GPS and GPS signal and calibration is not the goal of this work. The user can obtain any kind of data in the field with GIS and GPS mobile technology. The GPS is given 4 to 15 meter accuracy. However, as we also use satellite image as base layer, the accuracy of the map depends on the quality of this image. In this work we used a 5 meter resolution satellite image (as explained in result chapter). Coupled field survey and image interpolation, definitely increase the quality of our data.  We explained in the result chapter, which satellite image we have used for clarification of data collection. We can use any satellite images based of our budget. In another test, for an area we did not have satellite images and we used google image.

Commentato [S23]: Title not acceptable. Integrate with previous chapter.

This chapter is updated and renamed to Implementation: Page 5 to 8

We have merged with previous one. We want to highlight that our application, compared to others we mentioned in background has an advantage and we can also record element at risk related to each event. So the final database not only has the data on landslides but also GIS data on damage (element at risk).

Commentato [S24]: Confusing. The previous methodology treats landslides with characteristics (materials, type, and damage). Here element at risk. Merge all data type in same chapter.

This chapter is updated and renamed to Implementation: Page 5 to 8

We have merged accordingly.

Commentato [S25]: Do you mean land cover mapping? Otherwise you cite a company…

Yes. We mean landcover mapping by geoville as it is referenced too.

Commentato [S26]: Too vague, specify simply your aim.

We mean GIS can use spatial data layers to see the effects of parameters. An example is in Results section by query the database and see all landslides happened near forest and mainly damaged roads.

Commentato [S27]: Which is the difference between Open Source Geospatial Software and Open-source geospatial technology. Clarify.

They use different technologies to implement software. What we used in this paper are all technologies. Examples of software are those mentioned in the beginning of the chapter (UNEP, 2014; Geoville, 2016; USHAHIDI, 2015). Software is available and ready to use, while technology is something we need to do

some more programming to achieve what we need.  The difference of technology and software is out of the context for this paper. They are actually being used as synonym here for not having redundancy.

Commentato [S28]: Which DBS? Who is the owner?

We have updated:

p.9 line 7: Combination with database management systems (PostgreSQL, 2015; PostGIS, 2015; MySQL, 2015; UserCake, 2015)

Commentato [S29]: Why do you repeat so many times?

We have deleted.

Commentato [S30]: This is innovative. You have to dedicate more time then past experiences on classic landslide database… Your app treats with PhoneGap, linked to existing web development. By website I read "hybrid applications built with HTML, CSS and JavaScript". You should specify which link on web storages, simply to include in your methodology.

This is not a hybrid application, as we mentioned, it has 2 versions. One is offline and one is online (as can be seen in figure 5). The output of offline map is geojson-txt files which will be uploaded to the online version when internet is available. As we had to spend 8 hours or sometimes 2 days in the field without internet or very poor internet, hybrid applications are not advantageous for our work.

Page 9: This chapter is updated  with more details on Architecture.

Commentato [S31]: Which one? Both.

Commentato [S32]: Correct but redundant sentence

We have removed it accordingly.

Commentato [S33]: Describe which info by photo

We have updated the implementation chapter with more information on database. Figure 4 Page 8.

Commentato [S34]: User profiles

It offers more than user profiles for the application. By meaning user management, users can manage different things and admin can define different public/private pages and privileges for different users. The online application will not be loaded if a user does not log in, which we refer to as authentication.

Commentato [S35]: Not all components of architecture are explained. Consider them and introduce.

We added them (PHP and JQuery) and updated.

This chapter is updated  with more details on Architecture please refer to page 10.

Commentato [S36]: Why do you use prototype definition? The app will be updated, is not completed working or you need a piloting?

This application is already tested for a couple of times in the field, however, it is still a prototype because to use it widely, it needs to be completed, updated and supported in different areas. This is not the final

product. We have mentioned some problems we faced while using it in the last chapter. We also provided some new versions for other institutes and universities for their works and test.

Commentato [S37]: Explain which kind of combination.

As can be seen in figure 8, there are different base layers from different sources. We can have different base layers in the offline version. Openstreetmaps, satellite images, google maps or vector data, all can be added in advance to both offline and online versions. We updated the sentence for better clarification.

p.10 line 24: "Map with combination of multi-source base layer (OpenStreetMap, Satellite image, vector data can be seen in figure 8)"

Commentato [S38]: Not clear which analysis you intend

To convert the stored Geosjon to database and then to .shp file, we do some analysis. For example, we convert latitude and longitude to a geometry column in PostGIS. The querying in database is also another type of analysis. An example of result is in figure 14, showing number of landslides that caused damages to roads and so on. Calculating the area is another analysis, which can be easily done in PostGIS. They are mentioned in the result section.

Commentato [S39]: Editing events of landslides based on satellite image is not innovative. Field survey exists since a lot. You should mark the online-offline technology as real advantage on field. You did not describe the relations to update database in online-offline condition. You should describe how can be data collected be synchronized. Do users choose online-offline mode or is automatic upgrade based on bandwith? Figure 8 is not innovative, simply you edit a polygon on a raster image, what is new?

As a whole, editing landslide events on satellite image using a mobile device/application in the field itself is innovative. This application was tested in the field with ITC (Netherlands), Yale-NUS College (Singapore), University of Kathmandu and ICIMOD (Nepal), UNEP and Canton Vaud for forestry in Switzerland, and University in Tunisia. They were all interested to use this application and requested for an updated version. Due to our limited time, we have provided an updated version only to Singapore, Nepal, and Canton Vaud for testing. Recently, YaleNUS College tested the offline application and they provided their feedbacks and comments on the application.

As we mentioned before, we don't synchronize data automatically. Data are saved in offline as Geojson and then uploaded to online or directly added online in database. The admin user has to deal with updates and other things, as it is a normal task in all organizations working with data.

Figure 8 shows an example of online version. We explained all the tasks that both offline and online versions can do in this paragraph:

**P.10 line 24:"The offline component of ROOMA (Figure 6) contains the following modules: 1. Geolocation, 2. Map with combination of multi-source base layer (Openstreetmaps, Satellite Image, vector data) 3. Map drawer (Line, Polygon, Rectangle and Marker) 4. Satellite image as the base layer and 5. Saving options as Geojson-txt file in the offline mode. The mapping process is quick and easy; different features can be drawn on a map drawer after geolocation. Following, different satellite images as base layers assist for finding different objects on the map. However, the online component presents**

**more modules besides map and geolocation modules: 1.Saving online events directly to database, 2. Photo mapping, 3.Photo and event clustering, 4. User privileges 5. Data storage and analysis, 6. Import from/Export to Shape files."**

And then, we provided some photos to give the reader an idea of how they look like, for example different base layers (S37). We have used available technologies; therefore, we agree that editing a polygon itself is not innovative, but the application can be considered as innovative as a whole, especially as it provides an offline-online approach for data collection in the field. As you mentioned before, the architecture is very complex and to come up with this approach, we had to merge and program different functionalities of the application.

Commentato [S40]: Clarify distance, features, polygon revealed. Actually it is only a picture.

This chapter explains about the study area and therefore, figure 10 is only an overview of the area we did in our field survey. We clarified and mentioned the results achieved in the result section.

Commentato [S41]: Parameters not present in previous list within methodology. Did you add new text? Why?

These are not parameters, but an example of what could be added while recording landslides. To avoid redundancy and to give a clear idea, we added the "e.g." so that the reader can remember what we mean by land use features. We have updated it again as follows:

P.13 line 9: "The mapping of landslides (using polygons) was accompanied by data collection on land use features for each event (e.g. roads, rivers and forests)"

Commentato [S42]: Clarify the link between mobile-GIS and frequency distribution

Frequency is deleted.

P.13 line 10: "The advantage of mobile-GIS is increased in relation to the existence of landslides and distribution of landslide areas."

Commentato [S43]: It is a reason why you integrate satellite image. It has to be mentioned within definition of methodology

We have added accordingly.

P.6 line 2: "The satellite image added to the application significantly eased the exploration of this area and assisted the visual interpretation process."

Commentato [S44]: It is a bit ambiguous. You update landslides with you field actions, but some of events are not accessible, but visible only with distance like in Figure 12. I would consider as integration.

Yes, that is why we mentioned "assisted visual integration". You can easily look around and look at satellite image, and confirm landslides or not. We combined both field surveys with visual interpretation. We clearly mentioned that before (S43).

Commentato [S45]: Large landslides are visible on satellite. Did you edit on desktop GIS or check shape of landslide on field. IT could be a tool to upgrade what is existing as polygon.

We collected all landslides in the fields, and then compared with an example we did in the office after. (Figure 15) page 16.

Our work only took 2 days of field trip and one day of uploading them to online, and the work in office took couple of weeks. There was no existing polygon when we started the field trip. That is true having a tool to show existing polygon can be a good idea, but we do not have it in this work for offline version. As we mentioned, online version has the option to upload shp file.

Commentato [S46]: These are scientific and practical results. Not clear and too general. Delete.

We can have all different results and we have updated that in Abstract and Result that just some selected results are mentioned in this paper.

p.1 line 16: "This paper reviews the implementation and selected results of a secure mobile-map application called ROOMA (Rapid Offline-Online Mapping Application) for the fast data collection of landslide hazard and risk."

p.13 line 8: "we present some selected results. For example…."

Commentato [S47]: New version of output

Figure 3 is added for more clarification of output and workflow of this app. Please refer to new figure 3 in page 6, which is explain the output of this app.

In methodology, we mentioned that which type of data including materials and damages were gathered. (Refer to table 1). Thus for all the data we gathered, we can have different outputs. We can easily see distribution of landslides (figure 13) or we can also do query on the database and see distribution of landslides that are debris (materials) or can even see all the landslides which have high hazard degrees (Hazard factors) and select those areas as urgent areas to consider. We can make hazard susceptibility and by having element at risk, we can have risk. As we mentioned, we can have all different output maps for landslide hazard and risk, and we only mentioned some here.

Commentato [S48]: This conclusion is positive. You declare landslides in gullies visible with high resolution images, while adding filed survey you can define and edit better the polygons

This example was to emphasise that field survey even with a low resolution image can give us better view of the landslide. As in the field we simply noticed that is bigger landslide and not two separate landslides as drawn in the office. We have updated more clarification there.

Commentato [S49]: Reply all these aims within discussion. While you focus simply on data collection and database…

All work and articles need a conclusion at the end. Unfortunately we cannot delete conclusion from our paper. We updated it to "concluding remarks and discussion".

Commentato [S50]: Quite old sentence…

Commentato [S51]: You repeat several times elements at risk, but I do not see some examples about field trip or database about them

We have added the data model to the implementation (Figure 4 page 8).

 In our field trips, we did not collect and draw features for element at risks like roads or houses because there were crisis in Nepal those time and finding a Jeep with petrol was very hard so we had limited time using the jeep (2 days). However, we did gather them in our offline form of app ( Table 1: Number 8 and 9) by pointing out whether there are roads, houses or others close to this landslide or not and also whether they are damage or not. Figure 14 shows an example of damage data (Element at Risk which are damaged): how many landslides were close to the road and how many have damaged road.

Commentato [S52]: You do not mention before. Who participates and which organizations or teams.

We have mentioned that in

P.2 line 30 : " accessible to authorities, stakeholders and the general public"

P.18 line 18: "to use such data for landslide hazard and risk assessments for both stakeholders and local authorities)"

Data can be used by anyone who needs and are interested in it. They are so many organizations that are interested in landslide data. For example, transport companies need to see relations between roads and landslides and their damages.  Different participates and organization is out of context, we just mentioned it in overall.

Commentato [S53]: Concept of data synchronized and gain of time with ROOMA are not mentioned

As we mentioned already several times, there is no synchronization in this work. It was not necessary for us to include. Data for landslides can be updated from time to time. Thus, it will be added and updated to database based on the responsible admin's choice.

We did not do any scientific test for the Gain of time (if you meant to say fast acquiring with ROOMA, compared to other conventional approaches?)  But it is obvious collecting data in mobile is fast compare to paper work. Still most of the landslide field works collect information on the field with paper (We mentioned some in background) and those need to be added, typed, and drawn separately one by one in the office which require more times however we just need to click, upload file and the online application does all for us with one click.

**Review 2:**

"An Offline-Online WebGIS Android Application for Fast Data Acquisition of Landslide Hazard and Risk" deals with an interesting and innovative topic that is mobile tools for field landslide mapping. In particular the authors developed a prototypal App which enables the visualization of several cartographic satellite maps used as background layers upon which the user can draw the contours of the landslides recognized in the field. The App is able to upload data to a database once an internet connection is available and to export the products as shapefiles. The main issue with this paper is clarity. First of all the architecture of the system is not clear and a figure showing it is also missing. I suggest to better to explain the temporal process involved in the publication of the offline data. In particular what parts of process are automated and what are manual. Software with deferred updating may have problems when are used by more than one user. For example there may be problems due the digitization of the same landslide by two different users. What solutions have been adopted to solve this issue? English is often incorrect or not fluent to the point that only some of the major points have been signaled (see my specific remarks below).

Punctuation should also be revised. Therefore I recommend that the paper undergoes a professional English check.

Some figures also are often not completely clear or even contain errors. See the specific remarks below.

When citing more than one reference in the text be sure that they are sorted following the criteria of NHESS. All considered I recommend major revisions.

Dear Reviewer,

Thank you for your general comments. We would like to mention the following points regarding your general comments.

1. We have updated the technology part with the architecture. The online version uses the famous three tier architecture and the offline one is just an android app made using Cordova and phoneGap.

2. If people draw the same landslide, the application cannot recognize it. It just doesn't let you save it in the database if it has the same name because names are unique in database. So the admin has to check those in advance and if there is duplicate, delete one. Naming landslides when recording during different time is also something that should be carried by admin. For example, in our case we name the landslide after name of mapper_name of the area and maybe the year of the event:  RO_dhukurpokhari2015 so it is also easy to do the query later afterward. However this app was tested several times but it was not tested in one area more than once.

3. The English and reference correction were done.

4. We updated the figures as requested.

Following are the update and answer regarding your specific questions:

Specific remarks: Page 1 line 2: remove the semicolon.  Updated

P1 l14: what do you mean by "complications subject to accessibility and terrain"? One of the advantages of remote sensing is indeed to overcome accessibility issues.

RS are mostly used because of the accessibility and terrain. So of course that is an advantage of RS.

Updated as follow: "land-use mapping and hazard event inventories are mostly created by remote sensing data, subject to difficulties such as accessibility and terrain which need to be overcome."

P1 l16: add "the" before "implementation". Updated

P1 line 16: "This paper reviews the implementation.."

P1 l19: replace "for instance" with "such as". Updated

P1 line 20: "open-source web-GIS technologies such as Leaflet maps,"

P1 l20: remove "of". Also (here and elsewhere), PostgreSQL and PostGIS are cited like two separated product. It is better to report PostgreSQL as the real DBMS and PostGIS as its plugin for spatial database management.

Updated

P1 line 20: "This application comprises Leaflet map…"

P1 line 19: "This prototype assists the quick creation of landslide inventory maps (LIMs) by collecting information on the type, feature, volume, date and patterns of  landslides using open-source web-GIS technologies such as Leaflet maps, Cordova, GeoServer, PostgreSQL as the real DBMS (Database Management System) and Postgis as its plugin for spatial database management"

P2 l2, l7, l8, l9 etc: when citing references do not put an empty space before comma and do not use comma before "et al." (here and elsewhere in the text).

The references were added using REFRENCES in word 2010 authomaticaly. We have updated them all according to NHESS in the whole paper.

P2 line2: (Varnes, 1984)

P2 line7:  (Coe et al., 2004)

P2 line8: (Coe et al., 2004; Guzzetti et al., 2006; Hungr et al., 2014)

P2 l7: "selection of techniques relies on". Not clear, please rephrase.

Updated

P2 line 7: "however the selection of techniques depends on the size of the area, the resolution, the scale of the map, land-use, land-cover, soil and geomorphology (Coe et al., 2004; Guzzetti et al., 2006; Hungr et al., 2014)."

P2 l11: replace "are" with "is".

Updated

P2 line 10: "However, developing complete landslide inventories is difficult..."

P2 l15: replace "have long" with "require a long".

Updated

P2 line 14: "All methods for developing landslide inventories are resource intensive and time-consuming (Guzzetti et al., 2012)."

P2 l16: check English.

Updated

P2 line 15: "Landslides are often small, with a high frequency of occurrence and located in remote areas which are difficult to access."

P2 l18: replace "disadvantages" with "issues/problems". Updated

P2 line 17: "The lack of landslide documentation and databases is the main issue in the evaluation of landslide hazard risk; "

P3 l7: improvement with respect to what? Please clarify in the text.

Updated as follows:

P3 line7: "Data management improvement in hazard event mapping and storage using new technologies such as Postgis and Geoserver."

P3 l23: when references are cited within a sentence only the dates must be in the brackets.

Updated

P3 line 23: "There are many methodologies for landslide hazard assessment using geospatial technologies (van Westen, 1993; Soeters & van Westen, 1996; Guzzetti, 2000; Dai et al., 2002; van Westen et al., 2006)."

P4 l15: Another methodology that should be referenced in this paragraph is data mining from newspapers.

This figure related to techniques for landslide data acquisition and section has been removed from the paper as requested by the previous reviewer.

P5 l12: BGS Sigma is reported as 2013 in the reference list and further in the text.

Updated

P5 line 4: "The BGS digital field mapping system (BGS-SIGMA mobile 2013) includes…"

P6 l4-5: these two sentences are not connected with the following of the paragraph.

We have deleted those and we also updated the next chapter with figure 3.

P8 l29: the data transfer system between offline program and online component is not clear. Is it a normal web application? the data sending is automated or the user must select it manually from his device?

We have updated it. The transfer is done by uploading Geojson files from android app to the online version which mentioned several times in the paper. The online version is a normal web-mobile browser. The data are not being sent to the server automatically and the user has to upload those Geojson files from the device to the online system. The upload is available in the online version and can be done by one click.

P8 l35: It's not clear the role and the position of GeoServer and PostGIS DB (are they in a remote server? in the same server or in two separated servers? maybe a figure about System Architecture could help). Then it's not clear if GeoServer is used only as map server or if it is used also to receive the GeoJSON made from the mobile app (through native REST API or WFS protocol) or if this is done by another component connected to PostgreSQL/PostGIS.

Yes, our server was based in Geneva and we were working in Nepal. We used one server however, Postgis and GeoServer, which were installed separately in our system. Geoserver is used as map server and to export the final data into shpfiles. We have updated the technology section with more detailed information (Page9).

P9 l5: Is there a technical reason to use two different DBMS (MySQL and PostgreSQL) in the same project? (again a System Architecture figure could help).

Yes, we used MySQL for user management which was an open source package called UserCake, So we did not have to program and deal with user management separately and for spatial database we used Postgis which is under Postgres database.

As mentioned in the text:

P10 line 19: "UserCake library (UserCake, 2015) is an open-source library in PHP which using MySQL database (MySQL, 2015) to improve the user management and authentication"

P10 line 12: The FOSS4G technologies were selected to provide this module were PostgreSQL 9.4 (PostgreSQL, 2015) and Postgis 2.1 (PostGIS, 2015) for spatial database management.

P11 l7: replace "out team" with "the authors".

Updated.

P12 line 26: "where authors have been monitoring landslides since 2013"

P12 l7-8: this sentence it unclear. Please rephrase.

Updated:

P13 line 10: "The advantage of mobile-GIS is increased in relation to the existence of landslides and distribution of landslide areas."

P12 l14: in the centre of what?

In the center of our case study, Pokhara Lake Watershed. We have updated the text accordingly.

P14 line 4: "landslides occurred in the center of the Phewa Lake watershed."

P17 l3: here you have started the description of ROOMA concerning its database and then you talk about the test site. Before you start talking about the test site finish the description of ROOMA, talking about the offline drawing tool, the possibility to upload data and to export data in GIS format.

We have updated accordingly.

P17 l12-13: please be more conservative in this sentence, i.e. instead of stating that ROOMA will increase the quality of landslide maps, state that this it its aim, or that it provides a contribution in that direction.

Updated:

P18 line 13: "ROOMA tool aims to increase the quality ….. "

P17 l14-15: if this paper accomplishes something or not should be left to the reader to decide. Furthermore here you say that the paper accomplishes something that is still to be developed. Please change this sentence. Updated

P18 line 15: "This study can be improved through several of new developments to ROOMA, e.g. adding topographic data such as DEM and .."

Figure 2: what do you mean with "temporal"? The state of activity? Also, in the central box remove the capital letter from "L" in "Slope".

This refers to landslide maps for different time periods. It is a cited figure. We have updated "Slope".

Figure 3: this figure should be changed into a table. However in its present form it is very confused and confusing. Most notably, the second column should show names of techniques but also reports "frequency", "earthquakes" and others. Also, what are "exicting data"?

This figure is removed from the paper and it has been replaced with the following figure to give a better idea of how this application can work.

[Figure]

**Figure 3: Workflow of ROOMA where coupled image interpolation with field survey leads to asset of maps and complete database of landslide data and their characteristics. These different maps of landslide distribution, hazard, and damage infrastructure can be produced by manipulation in GIS.**

Figure 4: why in the landslide information are also here (left box) even though this concerns information concerning the elements at risk? Are not landslide information already contained in the landslide database (figure 3)? Please explain.

We have removed figure 3 and 4 as also requested by reviewer 1 and we updated it with a data model. Previous figure 4 was to show how the database could be used in our server by separating spatial and non-spatial data. We also updated the element at risk paragraph and added it to a previous caption with more details on database.

[Figure]

Figure 4: Data model of ROOMA: Database is automatically created from GeoJSON-text files which have been uploaded into online version of ROOMA.

Figure 10: in the caption replace "so many" with "several". Updated as requested

Figure 14: please add what is on the Y axis. Also the subdivision of a column between "feature" and "damage" is unclear. What do you mean by feature? This must be better explained.

The Y axis is number of Landslides, we have added it accordingly to the figure. Features are for each landslide, for example 56 landslides occurred in forests and of these, 43 damaged the forest (red = Damage). We also added more descriptions for clarity in the figure. We have updated the figure as follows:

[Figure]

**Figure 14: Relationship between features and landslides damage: for example 56 landslides occurred in forest and of these, 43 damaged the forest (Red = Damage).**

Table 1: It is not clear if the fields that you report here represent all the possible entries of your App or if there are just some reported as examples.

There are the possible entries for our app. We also added the data model in the new version for more clarity.

In the first case I suggest to add the actual interface of your App showing how filling in the landslide database works. Instead of "numbers of landslides" state "progressive identification number of the landslide". Also, why is it written "initiation" within the types of movements?

Filling the landslide database is conducted in mobile form (Table1). We will update the interface figure 6 by adding this form to the figure. We have removed initiation from types of movements

[revised manuscript text omitted]

| Landslide inventory | Hazard Factors | Element at risks |
|---|---|---|
| • Location, date, Type
• Depth, Volume
• Temporal | • Topographic factors like DEM , SLope, Aspect
• Enviromental factors
• Triggering factors | • Buildings
• Roads
• Population |

Historical landslide records and freely accessible databases have been developed for a few countries, (e.g. Italy (Guzzetti, 2000), Switzerland, France, Hong Kong (Ho, 2004), Canada and Colombia).  However, difficulties related to completeness in space and time are a drawback (van Westen et al., 2006).

| Landslide inventory | Hazard Factors | Element at risks |
|---|---|---|
| • Location, date, Type
• Depth, Volume
• Temporal | • Topographic factors like DEM , Slope, Aspect
• Enviromental factors
• Triggering factors | • Buildings
• Roads
• Population |

**Figure 2: Database for Landslide risk assessment and management (van Westen, 2004)**

**2.2  Techniques of Mobile and web GIS for landslide data collection**

10 Landslide inventories can be characterized by scale and the type of mapping (Guzzetti , et al., 2006). The different techniques for data collection are divided to: 1. Image interpretation 2. Semi-automated classification 3. Automated classification and 4. Field navigation including total stations, GPS and recently GIS mobile. Field works mostly are carried out to classify group of landslides triggered by an event, acquire data about characteristics of landslides, check inventory maps prepared by other methods, and improve visual interpretation of satellite images (van Westen, et al., 2006; Safaei, et al., 2010; van Westen , et

15 al., 2008). Figure 3 illustrates all the available techniques for the landslide data collection.

| Data: | • Techniques: |
|---|---|
| Satelite imagery | • Optical , Radar,Frequency |
| Airbone data | • Aerial photography , LiDAR, InSAR |
| Exicting data | • Geodesy , land use |
| Field data | • GPS,Total station, Mobile mapping |
| Labratory testing | • Soil, rock |
| Real time data | • Rainfall, earthquake |

Figure 3 : Overview of techniques for landslide data acquisition (van Westen, et al., 2006; Safaei, et al., 2010; van Westen , et al., 2008)

**2.3 Using GIS for landslide inventory**

5      Data obtained from field survey, laboratory, and image analysis can successfully been manipulated in the Open Source GIS and allow for graphics production, visualization, image processing, data management and spatial modelling. Many improvements in digital mapping and mobile GIS using Open-Source Geospatial technologies have been revealed in the field of data acquisition for landslide hazard and risk,. Followings are the examples of these technologies. The BGS digital field mapping system (BGS-SIGMA mobile 20122013) includes customisescustomised ArcMap 10 and Ms Access 2007. It is

10    designed which have customised two toolbars for mobile and desktop. The mobile toolbar was developed to capture the data in the field on rugged tablet PCs with integrated GPS units and requires Arc Editor Licence to run (BGS, 2013). Geodata implemented a mobile application that can add hazards as point markers with an attached image (GeoData, 2015). Another prototype for landslide geomorphological mapping using Geospatial Open Source source software such as MapServer and Postgis was implemented in the Olvera area, Spain (Mantovani, et al., 2010). (Mantovani et al., 2010). This application runs

15    on desktop and focuses more on data management system and visualization of data. WbLSIS (Acharya, et al., 2015) is Conceptual Framework(Acharya et al., 2015) is a desktop conceptual framework for Web-GIS Based Landslide Susceptibility. for Nepal with emphasis on data management. Another web-GIS tool was (Latini & Köbben, 2005) developed for landslide inventory using data driven SVG (Scalable Vector Graphics) and paper sketch maps with paper field works for landslide data collection. (Latini and Köbben, 2005). Temblor is a mobile application for the purpose of visualizing hazard maps online

20    anywhere (Temblor, 2016). And finallyLastly, Global disk platform by UNEP is a webWeb-GIS platform by usingwhich uses open-source canto visualize hazard maps and some other related data from so many countries (UNEP, 2014). Data but data available in that platform is limited. However there There are few workssystems with an option of using mobile technology for landslide and hazard field survey,surveys, while there are some other workssseveral related systems using satellite images and mobile GIS. For example, there is (e.g. a GIS mobile application (Bronder and Persson, 2013) for data collection of

25    cadastre (cadaster) mapping using EsriESRI and Google SDK (Bronder & Persson, 2013). Besides,). Geoville has developed

[revised manuscript text omitted]

---

## Referee Report (RR1)

25

[referee-annotated manuscript omitted]

---

## Author Response (AR2)

Dear Editor and Referee,

Many thanks for reviewing again this paper. Following you will find our answers and correction for the paper. We have updated most of the photos and we will also upload those figure separately for better quality.

The main changes are as follow:

    A.  All figures with small words are updated and now can be readable.
    B.  Figure 8 and 10 are updated.
    C.  Concluding remarks chapter is updated
    D.  The rest of updates are mentioned below.

1.  ROOMA was tested for rapid mapping of landslides in post-earthquake Nepal and can also be applied for all other events and hazards such as floods, avalanches, etc.

> rev
> This sentence appears only within conclusions chapters. It is not a follow up, better delete it.

    -   Answer:  Page 1 line 27:Deleted

2.  however the selection of techniques depends on the size of the area, the resolution, the scale of the map, land-use, land-cover, soil and geomorphology

> rev
> It does not appear within landlside data features you planned. So delete it…

    -   Answer: Page 2 line 7: This is just an introduction about how LIMs can be generated.  We have deleted Land-cover.

3.  Limited damage data are available for landslides, which is why developing landslide vulnerability assessments is challenging;

> rev
> Limited damages data are available for landslides, which are why developing landslide vulnerability assessments is challenging

    -   Answer: Updated:  Which are why developing landslide vulnerability….

4. Background

rev
·ú Landslide maps· is the keynote. You summarize risks assessment correctly. You have to highlight a bit more you focus within this task. Make a clear sentence about. The rest of risk concept concerning landslide is clear, but not your aim.

Answer: We mentioned about risk management to pinpoint that landslide inventories are one of the key element for risk and hazard management (Figure 1). Risk assessment is another key for risk management which we did not summarize in this chapter, because it is not our aim.

5. Includes customised ArcMap 10 and Ms Access 2007 which have customised 5 two toolbars for mobile and desktop.

rev
ArcMap 10 and Ms Access 2007 are not Open-Source Geospatial technologies!! Please clarify. Sentence not correct. Two toolbars about what?

- Answer: BGS-Sigma is free to use but to do that we need Arc map license. We mentioned this example here, because that was the most similar application to our application by using 2 version of mobile and desktop.
  We updated as follow:
  "Many improvements in digital mapping and mobile GIS using Geospatial technologies have been revealed in the field of data acquisition for landslide hazard and risk which mostly are open-source."
  Updated: Page 5 line 5 "includes customised ArcMap 10 and Ms Access 2007 which have customized two toolbars for mobile and desktop for digital geological mapping."

6. free software however, it requires Arc Editor License

Licence is not for free, neither open source. You combine with free software, so clarify better.

- Answer: The software is free to download and use, but to run it we need ArcMap license because it is based on ArcMap 10.

7. MapServer?

> rev
> Mapserver is an Open Source
> Geospatial Foundation, is an
> Open Source geographic data
> rendering engine. Use correct
> words, based on your aim·

Answer: Page 5 line 9:  Update:  "Another prototype for landslide geomorphological mapping using Open-source Geospatial Foundation software such as MapServer and Postgis"

8. This application runs on desktop and focuses more on data management system and visualization of data

> rev
> Which example, for which aim.
> Give a name and functions.

Answer: Page 5 line 10: Updated as: "Another prototype for landslide geomorphological mapping using Open-source Geospatial Foundation software such as MapServer and Postgis was implemented in the Olvera area, Spain to improve transportation and construction of roads (Mantovani et al., 2010)."

9. desktop conceptual framework …

> rev
> Which example, for which aim.
> Give a name and functions.

Answer: "WbLSIS (Acharya et al., 2015) is a desktop conceptual framework for Web-GIS Based Landslide Susceptibility for Nepal with emphasis on data management." Aim: Landslide susceptibility mapping.

10. Google SDK

> rev
> Android Studio?, Cloud SDK?, VR
> SDK? or Google API?. Sentence
> vague... You have to identify
> better  which kit you are listing

Answer: Page 5 line 19: Updated as: "Google Android SDK"

11. There are different systems in mobile GIS and data collection;

rev
Too vague sentence. Systems for landslides inventory?

Answer: we didn't mention landslide inventories, because as mentioned some of our examples are just using mobile GIS technologies and not related to landslide but they are used for data collection. So we wanted to emphasis, that while there are so many systems still we need a system that has satellite image in offline mode, precise mobile GPS, easy and fast drawing tools, advanced visualization, and database management system all in a one system.

12. visual interpretation process

rev
What does "visual interpretation" mean?

Answer: Visual interpretation: Visual elements of tone, shape, size, pattern, texture, shadow, and association (http://www.nrcan.gc.ca/node/9291)

13. GIS based maps including information such as landslide distribution, hazard, and damage infrastructure and a more complete database of landslide data and their characteristics.

rev
GIS based maps are products of spatial analysis after data survey, I suppose. You collect data geo-localized in offline concerning landslides. Why you mention specifically "distribution hazard and damages"?

Answer: our application and database has the information to create maps using hazard and damage infrastructure information. Figure 14 shows data on hazard and damage in a graph format. We can collect data on damage and hazard so we can have those maps too.

14. Inventorying a number of landslides and their spatial distribution is one method of creating landslide inventory.

rev
Landslides have spatial distributions. It is a common sentence. Delete it.

Answer: Page 6 line 29: Deleted

15. Android

rev
Android

Answer: Page 9 line 32: Update

16. Cordova and PhoneGap

rev
Explain how these technology offers field survey advantage without connection

Answer: Android apps are Standalone application. Cordova makes Android apps which are stand alone. That means you can run them without internet. Of course if your codes are in internet, then you need internet to run it.

17. Android environment.

rev
Explain why Android

Answer: Because we had Android tablets to run. Cordova can generate Android and IOS app at the same time using our JavaScript codes, we tested just Android version.

18. This enables the collection of data from multiple data collectors into the same database
rev
what?

Answer: We moved the sentence to one sentence up.

Page 10 line 8: "The data can be exported to GeoJSON-text files and uploaded through the internet to the online component where the main database is located. This enables the collection of data from multiple data collectors into the same database."

19. Geosjon

Updated: GeoJSON is a format for encoding a variety of geographic data structures which is similar to Keyhole Markup Language (KML) format, GeoJSON 2015

20. The FOSS4G technologies were selected to provide this module were PostgreSQL 9.4 (PostgreSQL, 2015) and Postgis 2.1 (PostGIS, 2015) for spatial database management.

rev
No sense. Rewrite

Answer: Updated as: Page 10 line 12
"The FOSS4G technologies selected for this module were PostgreSQL 9.4 (PostgreSQL, 2015) and Postgis 2.1 (PostGIS, 2015) for spatial database management.

21. The GeoServer 2.6 (GeoServer, 2015) module, in connection with Geodatabase (Postgis), is delivered for spatial analysis and visualization.

rev
If you mention spatial analysis
with Geoserver, please specify
which analysis

Answer: Spatial analysis are done in PostGIS that is why we mentioned GeoServer in connection with PostGIS. Querying in database is spatial analysis. An example of a simple query: "damage" LIKE '%% road' this query will just highlight all landslides which have damage on roads and then one can make a map of damage in roads and be used for reconstruction.

22. Geolocation

rev
You mean to geolocated a base layer or user geo-location during field survey? You do not speak about resolution of geolocation. In several other past tools (MapIt, ArcPad) fundamental is the accuracy of the location, especially if you are offline. Please clarify.

Answer: Updated to Geolocation using GPS on mobile or tablet.

Geolocation is done using mobile GPS, so accuracy is depend on GPS in the mobile. For our tablet we had a 4 meter accuracy. We can see accuracy in figure 6, 11 and 12. We will update better quality of those figures.  Geolocated a base layer? If you mean georeferenced, Base layers all are georeferenced before updating in application.

23. multi-source base layer (OpenStreetMap, Satellite image, vector data can be seen in figure 8)

rev
Advantages of the offline component are the survey in offline mode based on proper geolocation (GPS on mobile). How can base layers be added within screen?

Answer: We simply listed all the components in offline app not the advantages of application.

The base layer is added using leaflet library and to add satellite images, we need first to transfer them to tiles using TileMill and then add to leaflet codes.

Page 9 last line: "The satellite images are transferred to Tiles using TILEMILL (Mapbox, 2016) and added to Leaflet map library in both online and offline version."

24. Satellite image as the base layer …

rev
How can the satellite images be geolocalized? Fundamental for survey.

Answer: If you mean georeferenced, they are tiff files which already georeferenced, but in TILEMILL we need to do this transfer too:

*mb-util --scheme=wms roya2016.mbtiles roya2016*

*+proj=utm +zone=44 +datum=WGS84 +units=m +no_defs*

25. The ROOMA application was field tested for a rapid assessment of landslides triggered by this event or reactivated along with their land-use characteristics and damage to houses, schools, roads, rivers, agriculture fields and forest area (Figure 10).

rev
You focus on damages. Use a
more clear figure on them…

Answer: This photo is just an overall view of the area. We selected this photo because it is clearly showing so many landslide in this region which agriculture fields and roads were the most important damage. We can see landslides and damage to agriculture in this photo. We added some other photos to highlight damage agriculture and one landslide near a school.

Updated as follow:

[Figure]

Figure 10: A: Photo of the area with several landslides near Phewa LakePokhara watershed in Nepal and the damage to agriculture in blue circle, B: Photo of one landslide and damage to the road, C: Photo a house damaged by a landslide in same area

26. The advantage of mobile-GIS is increased in relation to the existence of landslides and distribution of landslide areas.

rev
No sense. Please clarify

Answer: We mean that mobile-GIS give an opportunity to see landslides that already existed and their distribution in that area. And it is continued by "The data were collected in the field using the offline version of this platform, either close to road or from a distance. This enabled easy interpretation of landslides which would have been difficult to access otherwise (Figure 11 and 12). Figure 11 represents a new landslide documented near the road that was not visible in satellite image and figure 12 shows a larger landslide which was located within a distance and clearly visible in image interpretation. Most of large landslides were mapped by distance. "To give better explanation of that sentence. We have updated as:

Page 14 line 18: "Mobile-GIS using satellite images and offline version gives an opportunity to see landslides that already existed and their distribution in that area."

27. Figure 13 shows the distribution of landslides in an area where most landslides occurred in the center of the Phewa Lake watershed.

rev
Figure 13 show me features about landslides distributed in a wathershed and view points of data gathering, obviously marginal and not always in good position. How can you complete these features with accuracy if your location is at distance

Answer: That is why we mentioned, we took advantage of Satellite images as base layer. We can see clearly the landslide (Figure 12). So quality of data is depend on the quality of satellite image and GPS.

28. All data were uploaded to the online version and then exported to a shape file, while the rest of the analysis was performed in QGIS2.6.1 (QGIS, 2015).

rev
I continue to not understand which kind of analysis you did before exporting...

Answer: Page 14 line 25 Updated as: All data were uploaded to the online version and then exported to a shape file, while preparing the maps (Figure 15) were performed in QGIS2.6.1 (QGIS, 2015).

29. Distribution

rev
I imagine classification of layers
using collected dataset, not
distribution…

Answer: We meant distribution of them. For example, we want to know which area has more damage. By simple query, we can see damages are mostly happened in the center.

30. Visualization of many data

rev
This is not analysis.

Answer: Sure. It is not. Deleted

31. For example, all the information about land-use characteristics and their damages for different landslide were gathered separately in our database and can be useful for more detailed analysis.

rev
Why gathered separately? You
have a db management system,
where did you save it?

Answer: We mean individually. That means we gather damage for each landslide and saved in different tables (Figure 4).
Updated to individually

32. Figure 12: Data collection by distance where was difficult to access however was easy to locate in the map using geolocation and satellite image

rev
If it is not easy to access, how
can you geolocate the landslide
on map in the background?

Answer: Using satellite image as base layer, we can see where we are and we can see the landslide and identify it based on it shapes (Figure 12).

33. Therefore, coupling satellite image interpretation with field observation allows one to identify better the type of landslide even using a medium resolution satellite image (~5 m).

rev
with accurate geolocation

Answer:  In this case (Distance) mostly satellite images helps to identify landslide not geolocation.

34. Figure 11: The map on the left shows the lower resolution image coupled with field survey and the map on right shows the same area with the detail mapping on standard GIS.

rev
I suggest another image. On the left a low resolution creates a general shape, and on the right you mark most accurate boundaries. The image is different with highest resolution, and more detailed drawing polygons. These features can be be done without field survey....
Make an example with landslide cover by vegatation or other not clear example...

I have some doubt that you complete shapes with that image accuracy available...

Answer:  Figure 11 shows an example when landslide is not visible in the map because it is a new landslide. But to identify landslides which covered by vegetation, figure 16 was the best example. The person in office identify that as 2 separate landslide however we identify it as 1 larger landslide

Update in page 18 line 1:  Figure 1:  The map on the left shows the lower resolution image coupled with field survey using offline application of ROOMA. The map on right shows the same area with the detail mapping on standard GIS. Field survey helped to understand that this is one larges landslide which is covered by vegetation however office work shows it as two separate landslides and ignored the part of landslide covered vegetation.

35. WebGIS technologies

rev
Which technologies you mean within your work?

Answer: Technologies that we mentioned in technology chapter.

36. The paper concludes that the ROOMA tool aims to increase the quality and speed of LIMs whether for susceptibility, hazard, risk assessments, and landscape modelling.

rev
Be vague and general cannot be your conclusion. Your work is data collecting on field, syncrhonized with a DB using online-offline. You do not treat any of these tasks, why do you list them?

Answer: We mentioned in background that landslide inventories are the basic for susceptibility, hazard, risk assessments. So improving the quality of LIMs will improve the quality of hazard and risk assessment.

Updated as follow: Page 19 line 13 "The paper concludes that the ROOMA tool aims to increase the quality and speed of LIMs which can improve the quality for susceptibility, hazard, risk assessments, and landscape modelling. "

37. Finally, it is essential to integrate a spatial decision support system to use such data for landslide hazard and risk assessments for both stakeholders and local authorities.

rev
Please list details of results, difficoulties, improvements and follow up, related only to your task.

**Details of results** is in result chapter. Results for this paper are:

1. Developing the Application
2. Testing the application
3. Comparison with another work.

Which all are mentioned in result chapter, we mentioned the results in summery in conclusion chapter here:

"This application incorporates rapid, economic and participatory methods for mapping landslides. It uses satellite images as multi-source map and enables multiple data collection to finally be collated in a centralized database. Data can be acquired in an offline version using an Android device or an online mode using all browsers in PCs, tablets and mobiles. The study was applied for mapping landslides in post-earthquake Nepal, but it can be applied for other hazard events such as floods, avalanches, etc. The result has been compared to the same study conducted remotely using image interpolation, and it shows that coupled field mapping with satellite image can improve the quality of landslide hazard and risk mapping."

**Difficulties** for landslide inventories are mentioned in chapter introduction however we didn't have any specific issues in the field except those mentioned in introduction which common in all works.

Chapter introduction:

1. **"**All methods for developing landslide inventories are resource intensive and time-consuming (Guzzetti et al., 2012).

2. Landslides are often small with high frequency of occurrence and located in remote areas which are difficult to access;

3. Landslides often have different characteristics which require them to be mapped and documented individually;

4. The lack of landslide documentation and databases is the main issue in the evaluation of landslide hazard risk;

5. Limited damage data are available for landslides, which are why developing landslide vulnerability assessments is challenging;

6. Sources of landslide inventories, such as aerial photography, satellite imagery, InSAR (Interferometric Synthetic Aperture Radar) and LiDAR (Light Detection and Ranging) are expensive. "

**Improvement** are mentioned in conclusion already:

1. Improvement for app: "The system is being further field tested for a future improved version; thus, this offline version can be improved by adding more components for distance calculation, continuous lines sketching, recording foot paths and merging the GPS located camera with the azimuth of data to help generate 3D models of the area."

2. Follow up studies: "This study can be improved through several of new developments to ROOMA, e.g. adding topographic data such as DEM and spatial-temporal modelling in order to increase accuracy. More effort is needed to incorporate and define vulnerability components, in order to generate risk maps. Finally, it is essential to integrate a spatial decision support system to use such data for landslide hazard and risk assessments for both stakeholders and local authorities."

We have added this list of results in conclusion chapter (Page 19 line 1) and mentioned we didn't have any specific issues except those mentioned in introduction such as difficulty to access to landslide. Please check page 19 line 1-20 for changes.

[revised manuscript text omitted]